JCB  Journal of Cell Biology

# Signaling by the integrated stress response kinase PKR is fine-tuned by dynamic clustering

Francesca Zappa[1], Nerea L. Muniozguren[1], Maxwell Z. Wilson[1], Michael S. Costello[1], Jose Carlos Ponce-Rojas[1], and Diego Acosta-Alvear[1]

The double-stranded RNA sensor kinase PKR is one of four integrated stress response (ISR) sensor kinases that phosphorylate the α subunit of eukaryotic initiation factor 2 (eIF2α) in response to stress. The current model of PKR activation considers the formation of back-to-back PKR dimers as a prerequisite for signal propagation. Here we show that PKR signaling involves the assembly of dynamic PKR clusters. PKR clustering is driven by ligand binding to PKR's sensor domain and by front-to-front interfaces between PKR's kinase domains. PKR clusters are discrete, heterogeneous, autonomous coalescences that share some protein components with processing bodies. Strikingly, eIF2α is not recruited to PKR clusters, and PKR cluster disruption enhances eIF2α phosphorylation. Together, these results support a model in which PKR clustering may limit encounters between PKR and eIF2α to buffer downstream signaling and prevent the ISR from misfiring.

## Introduction

The integrated stress response (ISR) is an evolutionarily conserved stress signaling network that adjusts cellular biosynthetic capacity according to need. Four stress sensor kinases govern mammalian ISR: GCN2, which detects uncharged tRNAs; heme-regulated inhibitor (HRI), which detects heme deficiency, redox imbalances, and acts as a signaling relay for mitochondrial stress; PKR-like ER kinase (PERK), which detects protein-folding perturbations in the lumen of the ER, or "ER stress"; and protein kinase RNA-activated (PKR), which detects double-stranded RNA (dsRNA). The ISR kinases phosphorylate the eukaryotic translation initiation factor eIF2—a heterotrimeric GTPase—on a single serine (Ser51) of its α subunit (eIF2α), causing a temporary shutdown of protein synthesis. Global translational repression by eIF2α phosphorylation is coupled to the selective synthesis of specific proteins, including the transcription factors ATF4 and CCAAT-enhancer-binding protein homologous protein (CHOP). Through this bipartite mechanism, the ISR reprograms the transcriptome and proteome (Costa-Mattioli and Walter, 2020).

PKR is the most recently evolved ISR kinase (Rothenburg et al., 2009). It has known roles in innate immunity (Pindel and Sadler, 2011; Cole, 2007) and in various neurological disorders characterized by cognitive decline (Peel, 2001; Bando et al., 2005; Hugon et al., 2017; Zhu et al., 2019). PKR detects viral and endogenous dsRNAs, including leaked mitochondrial transcripts, nuclear dsRNAs, and Alu-repeat RNAs (Ben-Asouli et al., 2002; Elbarbary et al., 2013; Kim et al., 2014; Youssef et al., 2015; Kim et al., 2018; Chung et al., 2018; Kim et al., 2020 Preprint; Lee et al., 2020a). Structurally, PKR is composed of two dsRNA binding domains (dsRBDs) and a kinase domain adjoined to the dsRBDs by a ~100–amino acid unstructured linker (Sadler and Williams, 2007). PKR forms back-to-back dimers sufficient for signal propagation upon activation (Maia de Oliveira et al., 2020; Dey et al., 2005; Lavoie et al., 2014; Cui et al., 2011; Dar et al., 2005). However, recent crystallographic evidence indicates that PKR could form high-order associations through front-to-front interfaces in PKR's kinase domain (Mayo et al., 2019). This observation suggests that PKR forms high-order associations in living cells, similar to the ER stress sensors PERK and IRE1 (Carrara et al., 2015; Korennykh et al., 2009; Li et al., 2010; Bertolotti et al., 2000; Cui et al., 2011; Belyy et al., 2021) and the innate immunity effector RNase L (Han et al., 2012). These independent lines of evidence hint at a conserved mechanistic principle of dynamic clustering of stress sensors upon activation.

To gain insights into PKR's activation mechanism, we used microscopy-based analyses to examine PKR's behavior in living cells. Our approaches revealed that upon activation, PKR assembles into autonomous, dynamic cytosolic clusters that are devoid of eIF2α, and that preventing PKR cluster formation enhanced PKR signaling. Taken together, our results highlight an unexpected feature of the ISR in which compartmentalization may modulate PKR-eIF2α interactions to fine-tune signaling.

[1]Department of Molecular, Cellular and Developmental Biology, University of California, Santa Barbara, Santa Barbara, CA.

Correspondence to Francesca Zappa: fzappa@ucsb.edu; Diego Acosta-Alvear: daa@lifesci.ucsb.edu

F. Zappa and D. Acosta-Alvear's present address is Altos Labs Bay Area Institute of Science, Altos Labs, Inc., Redwood City, CA.

**Rockefeller University Press**
J. Cell Biol. 2022 Vol. 221 No. 7 e202111100



## Results

### PKR forms dynamic clusters upon activation

To investigate the behavior of PKR in living cells, we introduced the red fluorescent protein mRuby in the interdomain linker of human PKR (Fig. S1 A). We used this construct to generate a stable H4 neuroglioma cell line expressing mRuby-PKR on the background of CRISPR interference (CRISPRi)-mediated knockdown of endogenous PKR. We chose H4 cells because maladaptive PKR signaling has been observed in several neuropathologies (Martinez et al., 2021). As expected, mRuby-PKR localized in the cytosol (Fig. S1 B). mRuby-PKR expression level was ~1.8-fold compared with endogenous PKR (Fig. S1 C), which was remarkably similar to the levels of endogenous PKR we observed in H4 cells stimulated with IFN, a natural PKR inducer (Fig. S1 D; Stark et al., 1998; Pindel and Sadler, 2011). The enzymatic activity of mRuby-PKR mirrored that of endogenous PKR, as determined by PKR autophosphorylation and eIF2α phosphorylation kinetics upon treatment with poly I:C, a synthetic dsRNA mimetic and potent PKR activator (Balachandran et al., 2000; Fig. S1 E). Poly I:C treatment also led to the formation of mRuby-PKR clusters within ~20 min (Fig. 1, A and B; and Video 1). The number of mRuby-PKR clusters per cell averaged 14.7 ± 7.93 (mean ± SEM), and their size ranged from 0.22 to 8 µm in diameter (Fig. S1 F). mRuby-PKR clusters coalesced and segregated within minutes, indicating dynamic behavior (Fig. 1 C and Video 1), and immunofluorescence analyses showed that mRuby-PKR in the clusters was phosphorylated (Fig. 1 D). Notably, endogenous PKR also assembled into clusters upon poly I:C treatment (Figs. 1 E and S1 G; Corbet et al., 2022 Preprint), indicating that mRuby-PKR recapitulates the behavior of the endogenous protein.

Next, we investigated whether mRuby-PKR would exhibit the same behavior in response to natural dsRNAs. PKR's best-known role is to detect viral dsRNAs, including those generated by the measles virus (MV; García et al., 2007). Infection of H4 cells expressing mRuby-PKR with a mutant MV strain (MV$^{CKO}$) that potently activates PKR (Okonski and Samuel, 2013; Toth et al., 2009; Pfaller et al., 2014) induced mRuby-PKR cluster assembly within ~38 h, which is consistent with the timeline of viral replication tracked by GFP signal (Pfaller et al., 2014; Fig. 1, F and G). PKR has also been shown to be activated by endogenous dsRNAs, including nuclear dsRNAs that are released into the cytosol upon disruption of the nuclear envelope during mitosis (Kim et al., 2014; Youssef et al., 2015; Kim et al., 2018). In agreement with these findings, we observed the formation of mRuby-PKR clusters in H4 cells undergoing mitosis (Fig. S1 H, arrowheads; Video 2).

The formation of PKR clusters suggests a form of compartmentalization that is evocative of biological coacervates (Fare et al., 2021). To address whether PKR clusters exhibit coacervate-like behavior, we took two complementary approaches. First, we treated cells in which we induced mRuby-PKR clustering with 1,6-hexanediol, a hydrophilic alcohol that dissolves coacervates (Kroschwald et al., 2015; Alberti et al., 2019). Treatment with 1,6-hexanediol dissolved mRuby-PKR clusters within ~15 min (Fig. 1 H). Second, to study the dynamics of mRuby-PKR clustering, we performed FRAP analyses.

FRAP analyses revealed a half-life of 3.93 ± 0.18 s between the cluster and cytosolic mRuby-PKR pools (Fig. S1, I and J). Notably, mRuby-PKR fluorescence in the clusters recovered to only ~50% of its initial intensity (Fig. 1 I), indicating that PKR clusters consist of at least two different PKR pools, one that quickly exchanges with the cytosol and another that is stably recruited into the cluster. Taken together, these results suggest that PKR signaling entails the formation of dynamic coalescences that are reminiscent of coacervates.

### PKR clusters are autonomous entities

It is not surprising that PKR, being an RNA-binding protein, has been observed in association with ribonucleoprotein (RNP) complexes such as processing bodies (PBs) and stress granules (SGs; Reineke and Lloyd, 2015; Reineke et al., 2015; Hebner et al., 2006; Dougherty et al., 2014), which are heterogeneous cytosolic liquid-like RNA granules that regulate mRNA metabolism (Protter and Parker, 2016; Luo et al., 2018). These observations suggest that PKR partitions to RNP complexes upon activation and during signaling. Poly I:C treatment of H4 cells expressing mRuby-PKR induced G3BP1 puncta, a canonical marker of SGs and RNase L-dependent bodies, a recently identified class of RNP complex (Burke et al., 2020; Fig. 2 A). However, mRuby-PKR clusters failed to colocalize with G3BP1 in these experimental conditions (Fig. 2 A). In line with these findings, PKR clusters and G3BP1 did not colocalize in the absence of RNase L-dependent bodies (Corbet et al., 2022 Preprint). Immunofluorescence analyses of poly I:C–treated H4 cells expressing mRuby-PKR indicated that most—but not all—mRuby-PKR clusters colocalized with Edc3, a canonical PB marker (Fig. S2 A). Moreover, we found an inverse correlation between PKR cluster size and Edc3 colocalization, wherein PKR clusters exceeding an average size of 2.50 ± 0.29 µm in diameter consistently failed to colocalize with Edc3 (Fig. S2 A). Imaging analyses in fixed cells coexpressing mRuby-PKR and GFP-Dcp1a, a fluorescently tagged canonical PB marker, showed the same colocalization pattern (Fig. S2 B). Thus, mRuby-PKR clusters colocalized in part with PBs but not SG markers.

To gain insight into the dynamics of PKR-PB associations, we performed superresolution live-imaging microscopy in cells expressing mRuby-PKR and GFP-Dcp1a (Fig. 2 B and Video 3). These experiments revealed that some mRuby-PKR clusters colocalize with GFP-Dcp1a upon poly I:C stimulation (Fig. 2 B), yet ejection of mRuby-PKR from GFP-Dcp1a–containing clusters occurred shortly after (~15 min after cluster assembly; Fig. 2 B and Video 3). However, mRuby-PKR and GFP-Dcp1a coalescences remained in apposition after de-mixing, indicating potential tethering (Fig. 2 B and Video 3). Furthermore, these experiments showed that ~50% of mRuby-PKR clusters consistently failed to colocalize with GFP-Dcp1a (Fig. 2 C), suggesting the existence of an autonomous pool of PKR clusters devoid of GFP-Dcp1a. The discrepancy between the extent of PB and PKR cluster colocalization obtained through superresolution live-cell imaging and conventional imaging carried out in fixed cells can be attributed to the effects of the fixative we used and further substantiates that PKR clusters are highly dynamic.

The interconnectivity between PKR clusters and PBs prompted us to investigate their potential interdependence. To

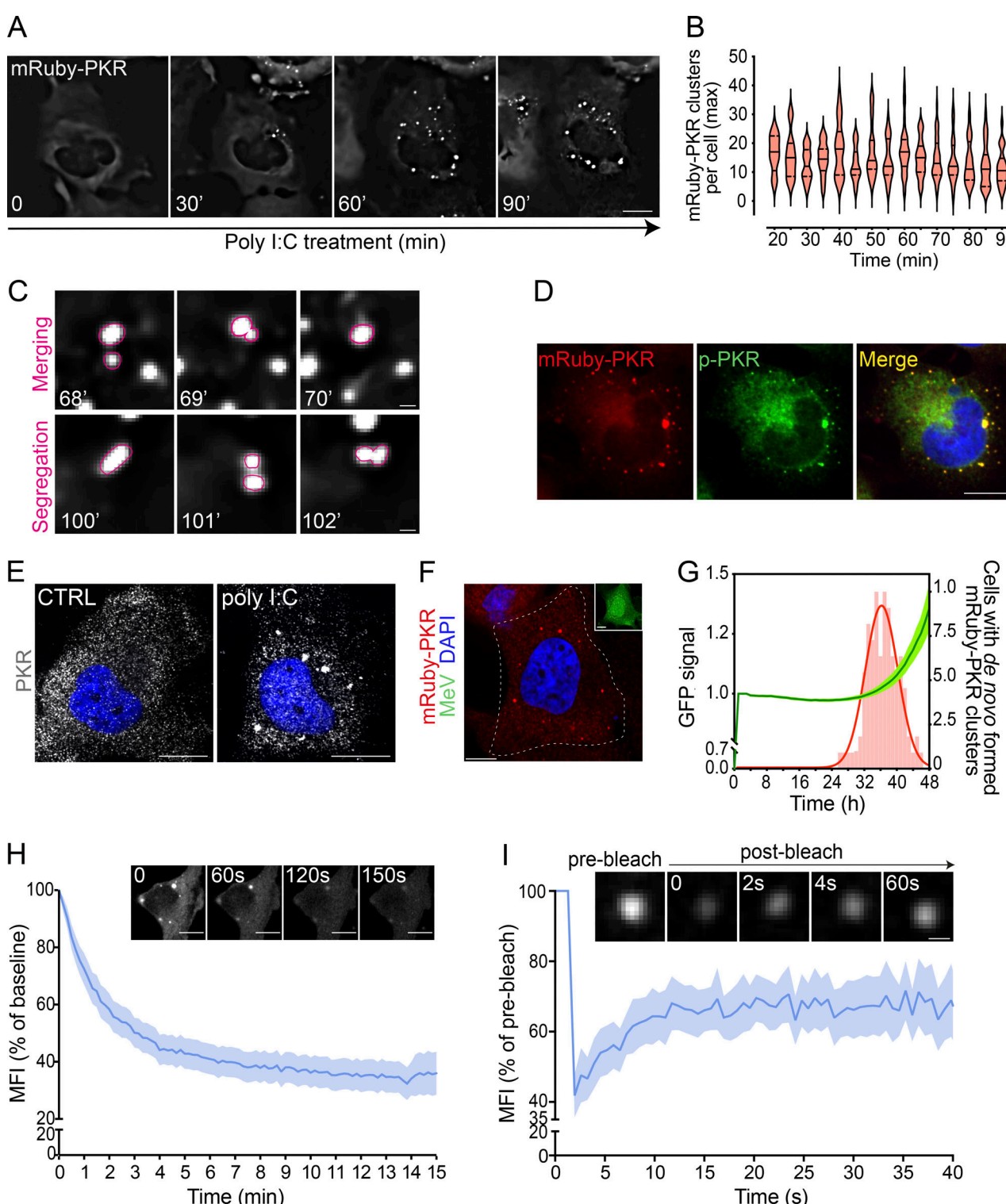

**Figure 1. mRuby-PKR forms clusters in response to synthetic and natural inputs. (A)** Representative time-lapse micrographs showing discrete mRuby-PKR clusters in H4 cells transfected with poly I:C. mRuby-PKR in clusters comprises 5.06% ± 0.2 of the total amount of protein (see Materials and Methods for details). Scale bar: 10 µm. **(B)** Violin plots showing the maximum number of mRuby-PKR clusters per cell in H4 cells transfected with poly I:C ($N$ = 5 experiments, $n$ > 200 cells). **(C)** Representative live-imaging time-lapse micrographs showing merging and segregation of mRuby-PKR clusters. Scale bar: 2 µm. **(D)** Representative immunofluorescence image showing that phosphorylated mRuby-PKR is enriched in clusters. Scale bar: 10 µm. **(E)** Representative immunofluorescence images of wild-type H4 cells treated with poly I:C and immunostained for endogenous PKR. **(F)** Representative micrograph showing formation of mRuby-PKR clusters in measles-infected (strain MVvac-CKO-GFP) H4 cells. The inset corresponds with the outlined cell in the high-magnification image. Note that the uninfected cell in the top left corner shows no mRuby-PKR clusters. Scale bar: 10 µm. **(G)** Quantification of de novo mRuby-PKR clustering frequency (red bars) and MVvac-CKO-GFP replication (GFP MFI and 95% confidence interval bands) as function of time. Red trace, nonlinear curve fit of the mRuby-PKR clustering frequency data. The percentage of infected cells that formed mRuby-PKR clusters was, on average, 45 ± 5% ($N$ = 3 experiments, $n$ > 400

cells). **(H)** Quantification of normalized mRuby-PKR cluster fluorescence intensity in 1,6-hexanediol–treated H4 cells. T0 corresponds to 60 min of poly I:C treatment. The data were binned and are shown as the mean and 95% confidence interval bands (n = 60 cells). The micrographs show representative images of mRuby-PKR cluster dissolution by 1,6-hexanediol treatment. Scale bar: 10 μm. **(I)** FRAP analysis showing the recovery of normalized fluorescence intensity of mRuby-PKR clusters. The data are shown as in H (N = 3 experiments, n = 30 cells). The micrographs show representative images of a single mRuby-PKR cluster photobleached with a 561-nm laser beam.

this end, we used two orthogonal approaches. First, we treated mRuby-PKR–expressing H4 cells with cycloheximide (CHX), a translation inhibitor that leads to polysome stabilization and depletion of PBs (Fig. S2, C and D; and Sheth and Parker, 2003; Cougot et al., 2004). Second, we knocked down 4E-T, the transporter of the mRNA cap-binding protein eIF4E, by RNAi, which also led to PB depletion (Fig. S2, C, D, and F; and Andrei et al., 2005). Neither CHX nor 4E-T RNAi hampered mRuby-PKR cluster assembly or affected their dynamic behavior upon poly I:C treatment (Fig. 2, D and E). Surprisingly, both CHX treatment and 4E-T RNAi led to the recruitment of Edc3 to mRuby-PKR clusters (Fig. 2 F), and CHX induced redistribution of 4E-T—which localizes to PBs—into mRuby-PKR clusters upon poly I:C administration (Fig. S2 E), suggesting that PKR clusters are capable of recruiting PB components.

Next, we investigated potential associations between PKR clusters and membrane-bound organelles. Immunofluorescence analyses in H4 cells expressing mRuby-PKR treated with poly I:C revealed that mRuby-PKR clusters did not associate with lysosomes, endosomes, autophagosomes, peroxisomes, or the cis-medial Golgi apparatus (Fig. S3 A). By contrast, live-cell imaging analyses revealed transient interactions with the mitochondrial network and the ER (Fig. S3 B; and Videos 4 and 5). These observations are consistent with recent reports suggesting that PKR localizes to mitochondria and that it senses mitochondrial transcripts (Kim et al., 2018; Lee et al., 2020a). Moreover, the transient associations between mRuby-PKR clusters and the ER align with recent findings showing that membrane-less organelles contact the ER (Ma and Mayr, 2018; Lee et al., 2020b).

To test whether PKR dimerization is sufficient to seed PKR clusters, we used a pharmacogenetics approach in which we replaced PKR's dsRBDs with a "bump-and-hole" mutant (F36V) of the FKBP binding protein, which dimerizes with the synthetic bivalent ligand AP20187 (Clackson et al., 1998; Yang et al., 2000; Fig. 3 A). As expected, treating cells expressing FKBP-PKR with the dimerizer led to FKBP-PKR and eIF2α phosphorylation, protein synthesis shutdown, and induction of canonical ISR markers, including ATF4 and CHOP (Fig. 3, B–E; and Fig. S4 A). As occurred with endogenous PKR and mRuby-PKR, activation of FKBP-PKR led to cluster formation (Fig. 3 F), which is consistent with a recent report indicating that PKR's kinase domains have front-to-front in addition to back-to-back interfaces (Mayo et al., 2019). FKBP-PKR clusters formed after 5 min of dimerizer treatment (Fig. 3 F), and they were smaller than mRuby-PKR clusters, averaging 0.12 μm in diameter. Unlike the more persistent mRuby-PKR clusters, FKBP-PKR clusters completely dissolved 60 min after the addition of the dimerizer (Fig. 3, F and G), suggesting that PKR's dsRBDs and RNA binding are required to stabilize the clusters. Accordingly, disrupting PKR's dsRNA binding capacity dramatically reduced its ability to cluster

(Corbet et al., 2022 Preprint). Despite these differences, FKBP-PKR clusters colocalized with PBs (Fig. S4 B), as occurred with some mRuby-PKR clusters (Fig. 2 B; and Fig. S2, A and B).

Last, we tested the ability of catalytically inactive mutant versions of mRuby-PKR and FKBP-PKR (K296R and T446A; Thomis and Samuel, 1993; Romano et al., 1998) to form clusters. Disruption of kinase activity did not impair cluster formation (Video 6) but modestly enhanced it (Figs. 3 H and S4 C; Corbet et al., 2022 Preprint). Taken together, these results indicate that ligand binding—RNA or dimerizer for PKR and FKBP-PKR, respectively—but not kinase activity, is required to nucleate PKR clusters.

### eIF2α is not recruited to PKR clusters

Besides itself, PKR's best-characterized substrate is eIF2α (Thomis and Samuel, 1993; Dey et al., 2005). PKR interacts with eIF2α through the C-terminal catalytic lobe of its kinase domain (Dar et al., 2005). In an active PKR dimer, these catalytic lobes face away from the back-to-back dimer-forming interfaces between the kinase domains, which allows each PKR protomer to interact with eIF2α in a 1:1 stoichiometric ratio (Dar et al., 2005). Our observation that PKR clusters are composed of static and mobile fractions (Fig. 1 I) implies that PKR clusters may limit the accessibility of eIF2α to active PKR pools. To investigate whether eIF2α enters PKR clusters, we conducted live-cell imaging analyses in H4 cells that coexpress mRuby-PKR and eIF2α fused to the green fluorescent protein mNeon. To our surprise, we found that mNeon-eIF2α was diffuse in the cytosol and not enriched in mRuby-PKR clusters in cells treated with poly I:C (Fig. 4 A and Video 7). Immunofluorescence analyses using a Ser51 phospho-eIF2α antibody in mRuby-PKR–expressing cells treated with poly I:C showed phospho-eIF2α decorating the periphery of the clusters, which corroborated our findings (Fig. 4 B). As expected, poly I:C treatment significantly increased the levels of phospho-eIF2α, indicating activation of the ISR (Fig. 4, B and C).

Because eIF2α is part of a trimeric complex composed of eIF2α, β, and γ subunits with a combined mass of ~125 kD (Beilsten-Edmands et al., 2015; Llácer et al., 2015), it is possible that steric effects preclude accommodation of eIF2α into the active site of each PKR molecule in the cluster. To test whether this is the case, we generated a stable cell line coexpressing mRuby-PKR and the vaccinia virus eIF2α homolog K3L fused to mNeon. K3L is a small, 88–amino acid protein that mimics the N-terminus of eIF2α and does not bind eIF2β and eIF2γ (Davies et al., 1992; Dar and Sicheri, 2002). Thus, we reasoned that this small PKR pseudosubstrate would not encounter the potential steric hindrance of eIF2. Indeed, we found that mNeon-K3L could access mRuby-PKR clusters, albeit with a lag time of ~10 min after their formation (Fig. 4 D and Video 8). Taken

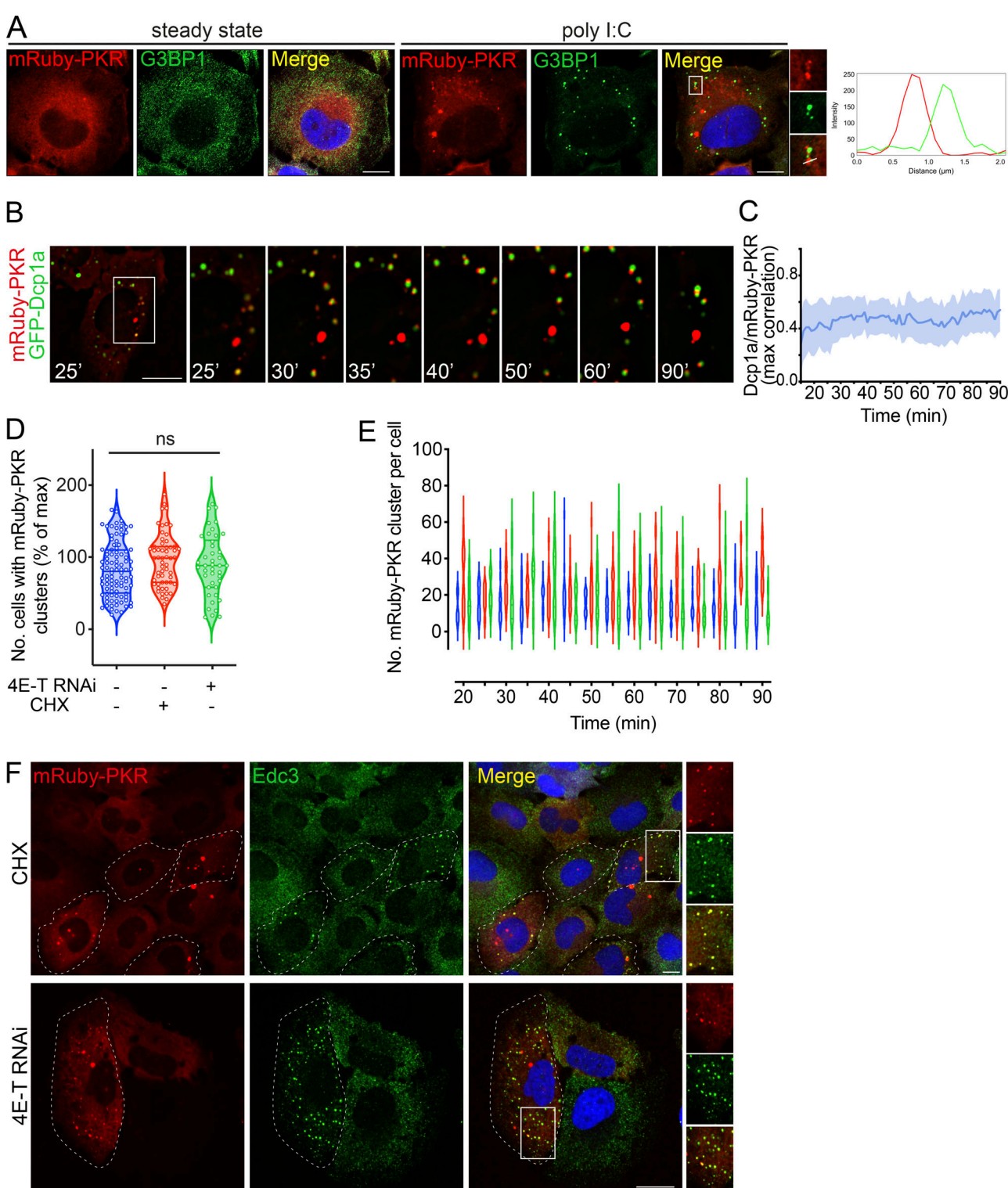

Figure 2. **PKR clusters are autonomous and recruit PB components. (A)** Representative immunofluorescence images showing that mRuby-PKR clusters and G3BP1, an SG component, do not colocalize. Right: Plot of signal intensity of mRuby-PKR clusters (red) or G3BP1 immunostaining (green) as a function of distance. The ROI used for metrics is indicated with a white line on the micrograph crops. Scale bars: 10 µm. **(B)** Time-lapse micrographs showing colocalization and subsequent demixing of mRuby-PKR and GFP-Dcp1a, a PB component. Scale bar: 10 µm. **(C)** Quantification of the data in B (mean and 95% confidence interval bands; $N = 3$ experiments, $n = 30$ cells). **(D)** Violin plots showing the total number of cells with mRuby-PKR clusters after administration of poly I:C (blue), poly I:C and CHX (red), and poly I:C and 4E-T RNAi (green); $N = 3$ experiments, $n > 2,000$; one-way ANOVA. **(E)** Violin plots showing the number of mRuby-PKR clusters per cell over time in cells treated with poly I:C (blue), poly I:C and CHX (red), and poly I:C and 4E-T RNAi (green); $N = 3$ experiments, $n > 200$. **(F)** Representative micrographs showing that mRuby-PKR clusters recruit the PB component Edc3 after depletion of PBs with CHX and 4E-T RNAi. Scale bar: 10 µm.

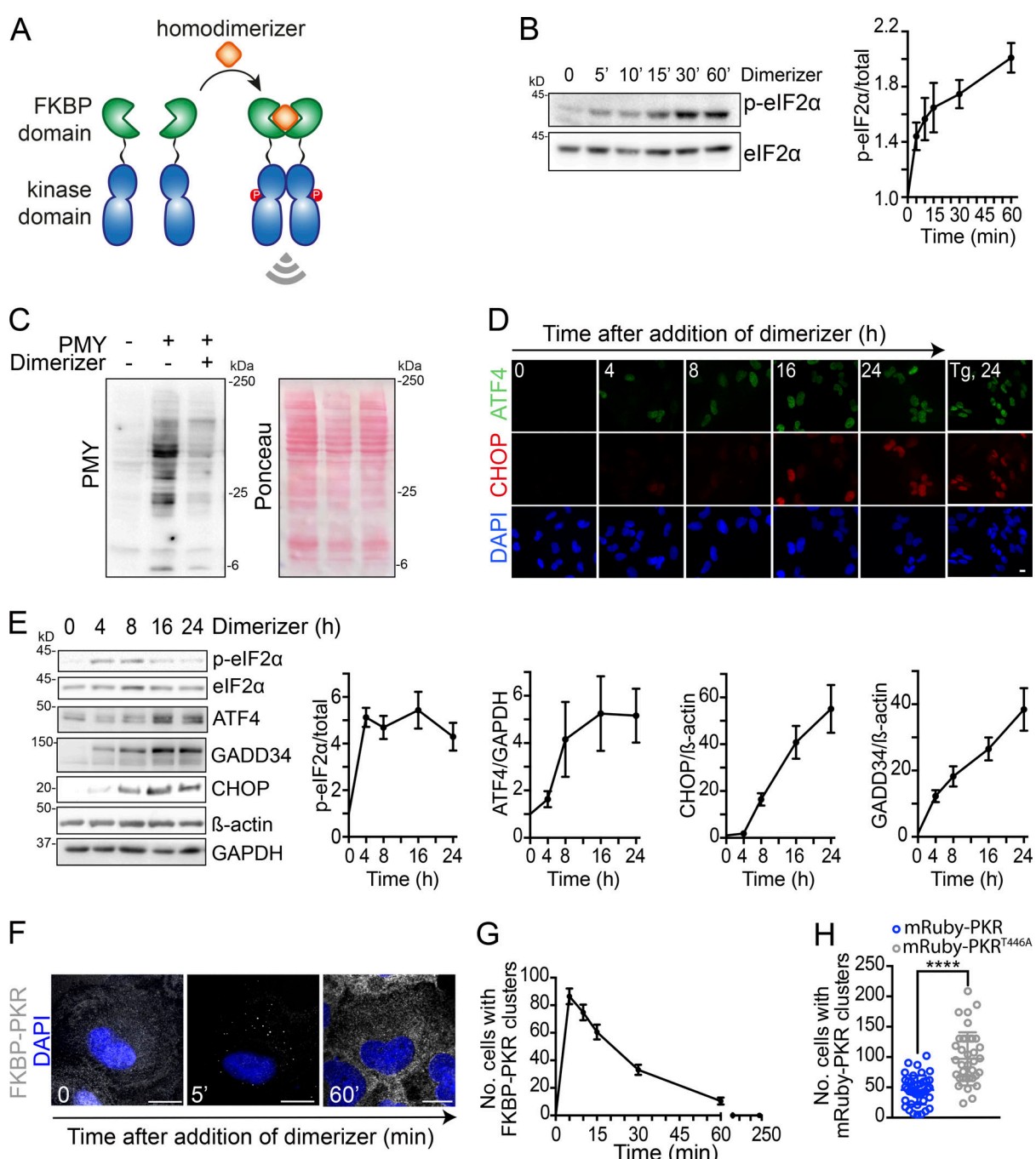

Figure 3. **PKR cluster formation requires ligand binding to PKR's sensor domain. (A)** Schematic representation of the pharmacogenetic approach for PKR activation using a synthetic dimerizer ligand. **(B)** Western blots showing forced dimerization of PKR results in phosphorylation of eIF2α. Right: Quantification of the extent of eIF2α phosphorylation (mean and SEM, N = 3 experiments). **(C)** Western blot showing forced-dimerization of PKR results in global protein synthesis shutdown as assessed by the abundance of puromycilated peptides. **(D)** Representative micrographs showing that forceddimerization of PKR results in accumulation of ATF4 and CHOP. Tg, 24, thapsigargin treatment (300 nM, 24 h; positive control). **(E)** Western blots showing forced-dimerization of PKR results in induction of canonical ISR target genes. β-Actin, GAPDH, loading controls. The right panels show the quantification of the data (mean and SEM, N = 3 experiments). **(F)** Representative micrograph showing that forced dimerization of PKR results in formation of PKR clusters. Scale bar: 10 μm. **(G)** Quantification of the data in panel F (mean and SEM); N = 3 experiments, n > 500. **(H)** Quantification of the number of cells with mRuby-PKR and catalytically inactive mRuby-PKR$^{T446A}$ clusters 90 min after poly I:C treatment (N = 3 experiments, n > 500; ****, P < 0.0001, unpaired Student's t test, nonparametric). Source data are available for this figure: SourceData F3.

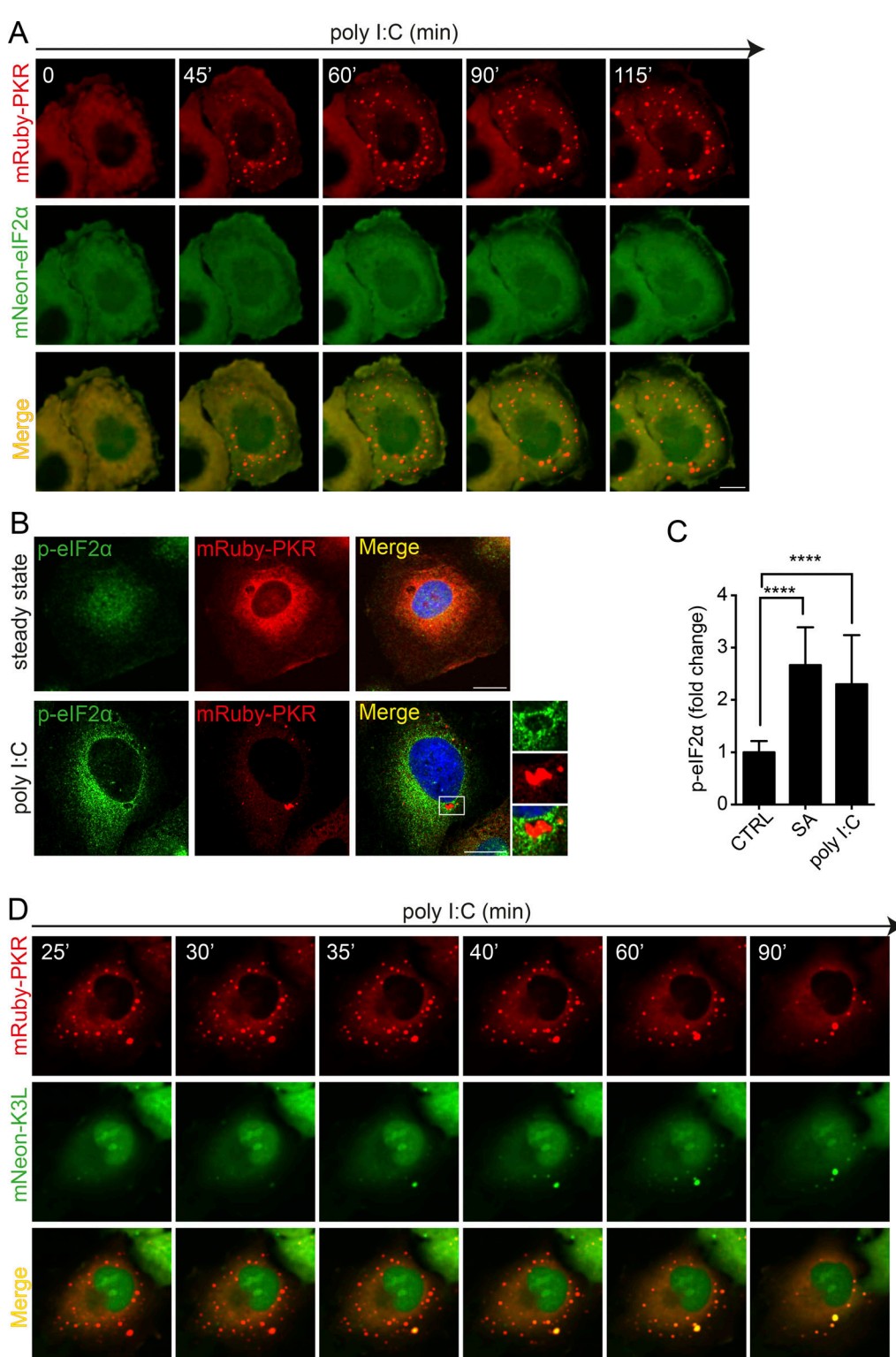

Figure 4. **eIF2α does not accumulate in PKR clusters. (A)** Representative time-lapse images of cells coexpressing mRuby-PKR and mNeon-eIF2α showing that mNeon-eIF2α does not accumulate in mRuby-PKR clusters in response to poly I:C treatment. **(B)** Representative immunofluorescence images showing that phosphorylated eIF2α is excluded from PKR clusters. The micrograph crops on the right show that phosphorylated eIF2α decorates the periphery of mRuby-PKR clusters. Scale bar: 10 µm. **(C)** Quantification of the MFI of the phosphorylated eIF2α signal in immunofluorescence analyses (mean fold-change and SEM, N = 3 experiments, n > 300; ****, P < 0.0001 unpaired Student's t test, nonparametric). SA, sodium arsenite, positive control. **(D)** Representative time-lapse images showing that the fluorescently labeled PKR pseudosubstrate mNeon-K3L enters mRuby-PKR clusters. Note the ~10-min time lag between formation of mRuby-PKR clusters and recruitment of mNeon-K3L to them. Scale bar: 10 µm.

together, these findings suggest that PKR clusters are unlikely the sites of eIF2α phosphorylation, but rather that they act as enzyme sinks that regulate the extent of eIF2α phosphorylation by limiting enzyme-substrate encounters.

## PKR cluster disruption accelerates and enhances eIF2α phosphorylation

To test the hypothesis that PKR clusters regulate eIF2α phosphorylation, we introduced mutations in PKR to disable clustering in vivo. We focused on two residues (S462 and G466) in PKR's kinase domain that have been recently shown to be required to stabilize front-to-front PKR kinase-domain interfaces in vitro (Mayo et al., 2019). We generated a stable H4 cell line expressing mRuby-PKR$^{S462A/G466L}$ in the background of CRISPRi-generated PKR depletion and tested the ability of this mRuby-PKR mutant to cluster upon poly I:C stimulation (Fig. 5 A). As expected, live-cell imaging analyses showed that the clustering ability of mRuby-PKR$^{S462A/G466L}$ was dramatically reduced (Figs. 5 B and S5 A; and Video 9).

Surprisingly, and in contrast to what has been reported in biochemical in vitro experiments (Mayo et al., 2019), disruption of PKR clustering in cells did not suppress PKR's self-phosphorylation but rather enhanced it (Fig. 5 G; and Fig. S5, B and C). Moreover, mRuby-PKR$^{S462A/G466L}$ expressing cells exposed to poly I:C showed accelerated and enhanced eIF2α phosphorylation that was coupled with increased ATF4 levels vs. cells expressing mRuby-PKR, even though the levels of mRuby-PKR$^{S462A/G466L}$ were lower than those of mRuby-PKR (Fig. 5, C and D; and Fig. S5, D and E). We observed similar results in cells expressing FKBP-PKR and FKBP-PKR$^{S462A/G466L}$, in which we activated signaling using the small molecule dimerizer (Fig. 5, E–H). These results suggest that the front-to-front interfaces in PKR's kinase domain promote cluster formation in cells, and that PKR clustering limits signal transduction, possibly by regulating PKR-eIF2α encounters.

## Discussion

Here, we identified a novel feature of PKR signaling: dynamic PKR clustering attenuates eIF2α phosphorylation, which we base on several lines of evidence. First, using live-cell imaging analyses, we show that mRuby-PKR reorganizes into visible clusters upon stimulation with synthetic, viral, and endogenous dsRNAs. Second, even though PKR clusters share components with PBs, we found that pharmacological and genetic ablation of PBs did not negatively impact PKR cluster assembly, indicating that clustering is an intrinsic property of PKR. Third, through mutagenesis analyses, we found that ligand-driven self-association and front-to-front PKR kinase interfaces, but not enzymatic activity, is required for cluster assembly. Fourth, our data indicate that eIF2α is excluded from PKR clusters, and disruption of PKR clustering enhanced downstream signaling. Taken together, our data support a model in which PKR clustering regulates enzyme-substrate interactions to potentially control the timing and amplitude of signaling.

The current model for PKR activation proposes that dsRNA-binding drives dimerization and trans-autophosphorylation to initiate signaling (Lemaire et al., 2005; Dar et al., 2005). This model is remarkably similar to the ER-resident stress sensor kinases IRE1 and PERK, which are also activated by self-association and trans-autophosphorylation (Zhou et al., 2006; Korennykh et al., 2009; Cui et al., 2011; Carrara et al., 2015). IRE1 and PERK form dynamic high-order oligomers, as does the pseudokinase RNase L, a key player in the antiviral response (Bertolotti et al., 2000; Lee et al., 2008; Korennykh et al., 2009; Li et al., 2010; Han et al., 2012; Carrara et al., 2015). We found that PKR exhibits the same tendency to form dynamic clusters upon activation. Moreover, the structural similarities between the kinase domains of the mammalian ISR sensors raise the possibility that clustering is pervasive among them (Taniuchi et al., 2016). These observations support the notion that stress sensor clustering may be a common organizing principle for signaling.

Our finding that PB components (e.g., Edc3, Dcp1a, 4E-T) are recruited to PKR clusters (Fig. 2 F) and a recent report showing that PKR-containing clusters include additional dsRNA binding proteins (Corbet et al., 2022 Preprint) indicate compositional heterogeneity of these entities. Notably, FKBP-PKR clusters recruit Edc3, which suggests that either the interlinker domain, the kinase domain, or both are required for interaction with PB components. In contrast to enduring mRuby-PKR clusters, FKBP-PKR clusters are short-lived (Fig. 3, F and G), indicating that PKR's dsRBDs—and RNA binding—stabilize PKR clusters and could further contribute to fine-tuning PKR signaling. It is noteworthy that the dissolution of PBs with CHX or upon genetic depletion of 4E-T does not influence the kinetics and efficiency of PKR cluster formation. In line with these findings, we observed PKR clusters form during mitosis—when PBs naturally dissolve (Yang et al., 2004)—indicating that PBs are dispensable for PKR clustering. These observations substantiate that PKR clusters are autonomous entities capable of recruiting PB components, potentially through a piggyback mechanism. Such a mechanism requires further investigations. Moreover, even though PKR has been found in SGs (Reineke and Lloyd, 2015), our analyses and those of Corbet et al. (2022; Preprint) indicate that PKR clusters and SGs are distinct. PKR clusters neither co-localize with SGs (Fig. 2 A) nor are they ablated by CHX (Fig. 2, D and E), which prevents SG formation (Mollet et al., 2008), which further attests to their autonomous nature.

Besides clustering, subcellular partitioning may provide an additional regulatory layer to control the ISR. Indeed, PERK is an ER-localized transmembrane protein (Harding et al., 1999), GCN2 associates with ribosomes (Harding et al., 2019), and HRI has been reported to act as a signaling relay for mitochondrial stress (Guo et al., 2020; Fessler et al., 2020). Likewise, PKR has been observed in the nucleus and in contact with the mitochondrial network (Kim et al., 2018; Jeffrey et al., 1995; Blalock et al., 2014; Fig. S2 B and Video 4). We also found that PKR clusters transiently associate with the ER (Fig. S2 B and Video 5), which raises the possibility that reshuffling PKR to different subcellular locales may regulate its access to local pools of eIF2. As a corollary, it is tempting to speculate that clustering of each ISR kinase in different subcellular niches could potentially control unique outputs.

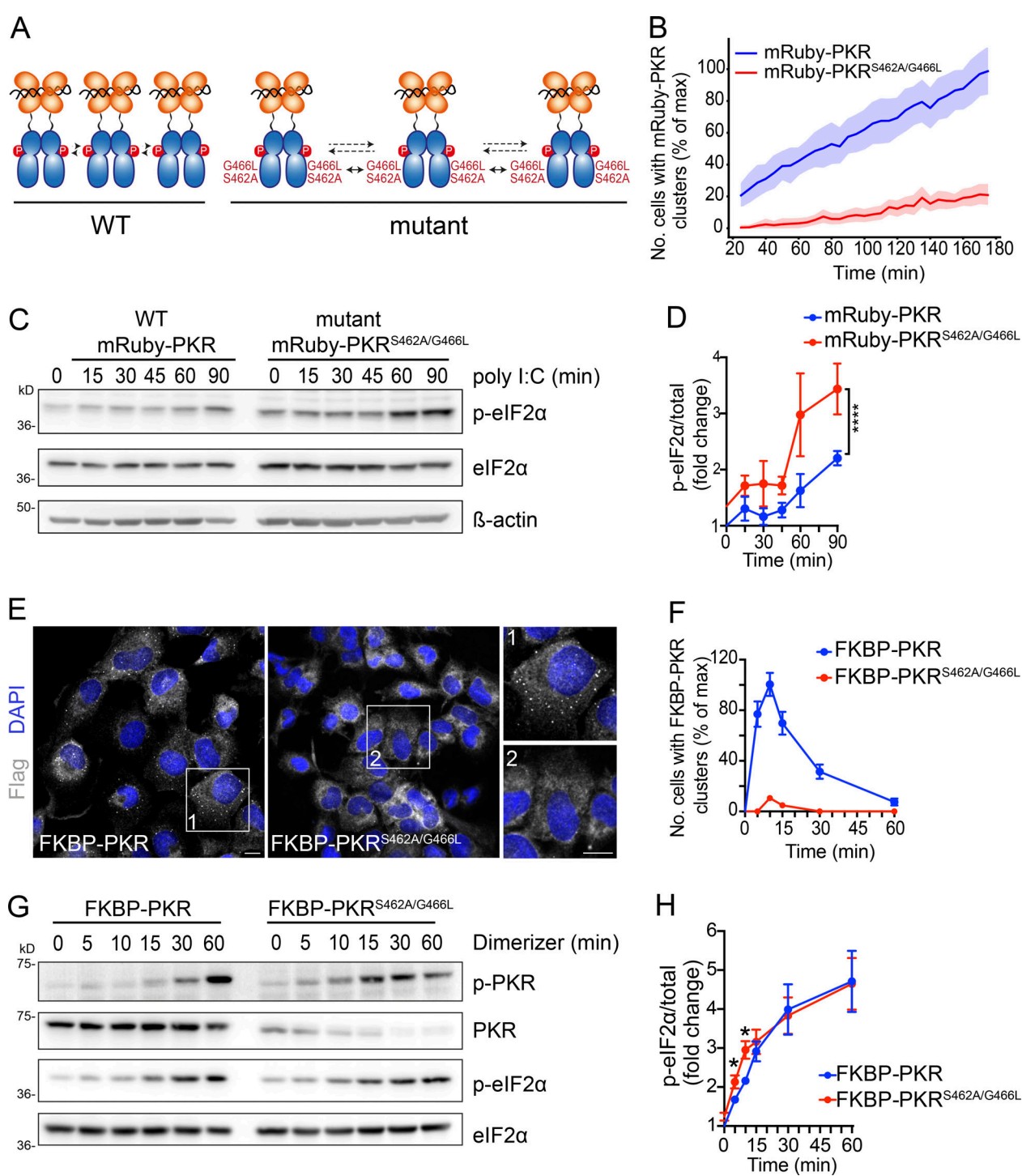

Figure 5. **PKR cluster disruption accelerates and enhances eIF2α phosphorylation. (A)** Schematic representation showing the mutations that disrupt PKR's front-to-front (FTF) kinase interfaces. **(B)** Quantification of imaging data showing that the mutations in PKR's FTF kinase interfaces severely reduce mRuby-PKR clusters in cells. Cells with <3 or >50 clusters were not considered in this analysis. The data were binned and are shown as the mean and 95% confidence interval bands; $n$ > 2,000. **(C)** Western blots showing that cluster-disrupting mutations in mRuby-PKR accelerate and enhance eIF2α phosphorylation in response to poly I:C treatment. **(D)** Quantification of the data in C (mean and SEM, $N$ = 3 experiments; ****, P < 0.0001, one-way ANOVA). **(E)** Representative immunofluorescence images showing that the mutations in PKR's FTF kinase interfaces impair FKBP-PKR cluster formation upon forced activation with a synthetic dimerizer. Scale bar: 10 μm. **(F)** Quantification of the data in D (mean and SEM, $N$ = 3 experiments, $n$ > 1,000; unpaired Student's $t$ test, nonparametric). **(G)** Western blots showing accelerated and enhanced FKBP-PKR autophosphorylation and phosphorylation of eIF2α upon mutation of PKR's FTF interfaces. **(H)** Quantification of the data in G (mean and SEM, $N$ = 3 experiments; *, P < 0.05; unpaired Student's $t$ test, nonparametric). Note that the augmented eIF2α phosphorylation is lost after 15 min, which is consistent with the time of dissolution of FKBP-PKR clusters (see Fig. 3 G). Source data are available for this figure: SourceData F5.

Our most intriguing observation is that eIF2α is excluded from PKR clusters (Fig. 4, A and B; and Video 7). Spatial reorganization can increase reaction rates of enzymatic reactions by concentrating enzymes and substrates into coalescences (Kohnhorst et al., 2017; An et al., 2008; Kim et al., 2019; Sheu-Gruttadauria and MacRae, 2018). However, it is difficult to reconcile our observations with this concept of "enzymatic factories," since disruption of PKR clusters in cells led to enhanced PKR autophosphorylation (Fig. 5 G; and Fig. S5, C and D), coupled with accelerated and boosted eIF2α phosphorylation (Fig. 5, C, D, G, and H). One possible explanation for this observation is that PKR's phosphatases can be recruited to the clusters to suppress excessive PKR signaling. Further experiments will be required to test this hypothesis.

Our results suggest that tight packing of active PKR dimers results in steric effects that preclude eIF2α from entering PKR clusters. Our observations with the small PKR pseudosubstrate K3L, which can access PKR clusters, lend support to this notion. Moreover, the ~10-min lag time between PKR cluster formation and K3L recruitment (Fig. 4 D and Video 8) suggests that cluster formation is not necessarily coupled with eIF2α phosphorylation. Given that PKR clusters appear to be biophysically heterogeneous (i.e., composed of a pool of PKR molecules that readily exchanges with the cytosol and another one that does not; Fig. 1 I), it is possible that active PKR dimers on the periphery of the cluster freely exchange with the cytosol to fine-tune enzyme–substrate interactions. A recent report indicating that IRE1 and its substrate, the XBP1 mRNA, do not meet in high-order assemblies in mammalian cells, but rather that the IRE1-driven splicing reaction occurs in diffuse ER locales (Gomez-Puerta et al., 2021), is consistent with our observations.

Based on the evidence collected, we propose a hierarchical PKR cluster assembly model (Fig. 6). In this model, dsRNA binding nucleates the formation of back-to-back PKR dimers. Indeed, a point mutation that abrogates dsRNA-binding reduces PKR's ability to cluster (Corbet et al., 2022 Preprint). PKR cluster biogenesis results from interactions among PKR dimers, which are facilitated by front-to-front interfaces in PKR's kinase domain. Such interactions lead to the coalescence of PKR dimers into higher-order assemblies. Heterologous protein–protein and protein–RNA interactions could further stabilize PKR clusters, for example, (i) upon recruitment of additional RNAs and RBPs, as recently described (Corbet et al., 2022 Preprint), (ii) as would occur when PKR clusters and PBs merge (Fig. 2 B), (iii) when PKR clusters decorate SGs (Fig. 2 A), or (iv) when PKR clusters encounter organelles (Fig. S3 B). While it is possible that PKR clustering constitutes an initial step in PKR activation (Corbet et al., 2022 Preprint), we interpret PKR clusters as molecular sinks that fine-tune the propagation of ISR signals by sequestering active PKR dimers from the bulk cytosol. In our model, the dynamic equilibrium between PKR molecules inside and outside the cluster enables active PKR dimers freed from the cluster to encounter eIF2α. Our findings support the intriguing possibility that stress sensor clustering calibrates the ISR to ensure nonadaptive outputs do not supersede homeostatic ones, thereby safeguarding the integrity of cells and tissues.

## Materials and methods
### Plasmid construction and generation of stable cell lines
H4 neuroglioma cells stably expressing a catalytically dead version of Cas9 (dCas9) fused to the KRAB transcriptional repressor domain were a kind gift of Martin Kampmann (University of California, San Francisco, San Francisco, CA). PKR was depleted in H4-dCas9-KRAB cells using CRISPRi as previously described (Gilbert et al., 2014). Briefly, H4 cells stably expressing dCas9-KRAB were transduced with a pool of lentiviruses encoding five different sgRNAs (5′-CCACCTTGTTGGGCCGCCGGC CGGAGACCCGGTTTAAGAGCTAAGCTG-3′; 5′-CCACCTTGTTGG GCGGCGGCGCAGGTGAGCAGTTTAAGAGCTAAGCTG-3′; 5′-CCA CCTTGTTGGGAAGCCGCGGGTCTCCGGCGTTTAAGAGCTAAG CTG-3′; 5′-CCACCTTGTTGGGGAAGACGAATAGGCCTAGGTTTA AGAGCTAAGCTG-3′; and 5′-CCACCTTGTTGGGGTCTAGTGGAA GACGAATGTTTAAGAGCTAAGCTG-3′) whose expression is driven by a human U6 promoter. The sgRNA sequences were obtained from the human genome-scale CRISPRi library developed by the laboratory of Jonathan Weissman. Cells expressing the sgRNAs were selected by treatment with puromycin (1 µg/ml) followed by FACS gating on blue fluorescence signal (the lentivector encoding sgRNAs also encodes BFP). mRuby-PKR was generated by in-fusion cloning of the PCR-amplified coding sequence of mRuby into the human PKR expression construct pDAA-002. pDAA-002 encodes a C-terminal FLAG-tagged version of human PKR hosted in the retroviral expression vector pLPCX (Clontech) and was generated by cloning a PCR product (PCRP) encoding the PKR coding sequence obtained from HEK-293 cell cDNA. This PCRP was obtained using oligonucleotides containing a 5′ HindIII site and a 3′ FLAG-epitope coding sequence and a NotI site and was cloned into the cognate sites of pLPCX using standard molecular biology techniques. FKBP-PKR was generated by cloning a PCRP encoding residues 170–551 of PKR of human origin obtained using oligonucleotides containing a 5′-BamHI site and a 3′ FLAG-epitope coding sequence and MfeI sites into the cognate sites of p1XDmrB-mCh-LRP6c (kind gift of Peter Walter, University of California, San Francisco, San Francisco, CA-Howard Hughes Medical Institute [HHMI]). The resulting construct, pDAA-006, replaces the mCh-LRP6c coding sequences in p1XDmrB-mCh-LRP6c with the above PKR coding sequence. The FLAG-epitope–tagged FKBP-PKR coding sequence was excised from pDAA-006 with XhoI and MfeI and subcloned into the XhoI and EcoRI sites of pLPCX-IRES-eGFP. pLPCX-IRES-eGFP was generated by cloning a fusion PCRP consisting of the encephalomyocarditis virus internal ribosomal entry site upstream of the eGFP coding sequence flanked by EcoRI and NotI sites into the cognate sites of pLPCX (Clontech). A DNA gene block encoding the vaccinia virus Wisconsin strain K3L fused to the C-terminus of mNeonGreen by a GSGS linker and hosted into the expression vector pTwist Lenti SFFV puro WPRE was obtained commercially (Twist Bioscience). eIF2α-mNeon was generated by in-fusion cloning of a PCRP encoding the mNeon coding sequence into the mouse eIF2α expression construct pDAA-026 to generate pDAA-025. pDAA-026 was generated by subcloning a DNA fragment encoding an N-terminus FLAG-tagged mouse eIF2α coding sequence flanked by BamHI and EcoRI sites into the BglII

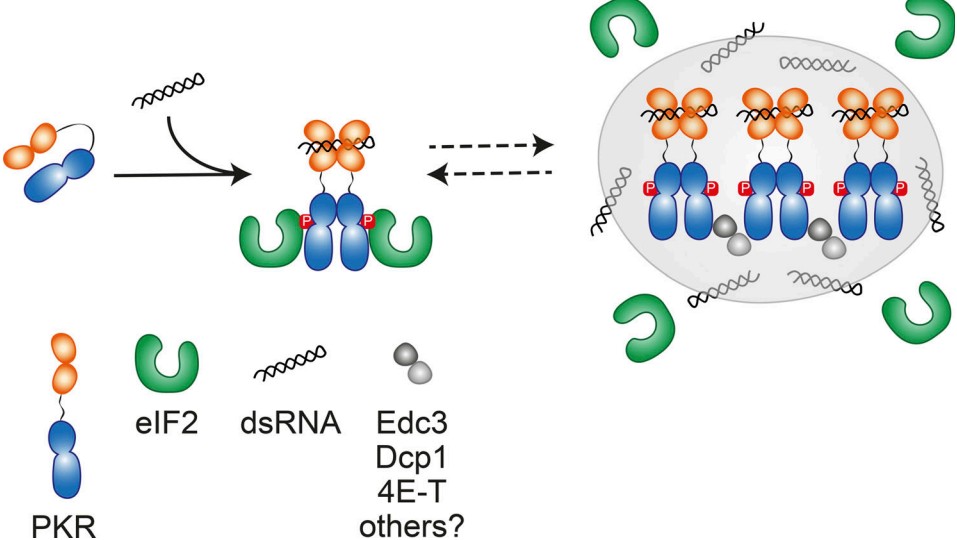

**Figure 6.** **Model for the assembly of PKR clusters and their role in fine-tuning signaling.** PKR is monomeric when inactive. dsRNA binding to its sensor domain drives dimerization and cluster assembly alongside PB components. The newly minted PKR clusters act as enzyme sinks that limit eIF2α phosphorylation, whereas cytosolic active PKR dimers that exchange with the clusters drive eIF2α phosphorylation.

and EcoRI sites of pLPCX (Clontech) using standard molecular biology methods. The coding sequence of wild-type FLAG-tagged eIF2α of mouse origin was obtained from a mammalian expression construct (Sidrauski et al., 2013). Point mutants of the PKR coding sequence were generated by site-directed mutagenesis of the corresponding expression constructs. The plasmid encoding GFP-Dcp1a was a kind gift of Gia Voeltz (University of Colorado, Boulder, Boulder, CO-HHMI; plasmid 153972; Addgene). The expression construct for ERmoxGFP was a kind gift of Erik Snapp (Janelia research campus-HHMI-Albert Einstein College of Medicine, Bronx, NY; plasmid 68072; Addgene). All viral vectors were used to generate recombinant lenti- and retroviruses and transduce cells as previously described (Sidrauski et al., 2013). Pseudoclonal stable cell lines were generated by FACS, selecting for a narrow gate encompassing the population expressing the midpoint level based on the signal intensity of the fluorescent reporters. This population typically comprised ~5% of the transductant population. Expression levels were maintained by treating the cells with puromycin (1 µg/ml). Whenever dark cell lines were generated, they were selected using puromycin (1 µg/ml).

**Cell culture, transfection, and drug treatments**

H4 cells were maintained in DMEM supplemented with 10% FBS, L-glutamine, and penicillin/streptomycin at 37°C, 5% $CO_2$ in a humidified incubator. Mixed molecular weight poly I:C (Tocris) was used at a final concentration of 2 µg/ml and transfected with Lipofectamine 2000 (Invitrogen) using the manufacturer's protocol. GFP-Dcp1a and ERmox-GFP transfections were carried out on 5 × 10^4 H4 cells in glass-bottom 24-well plates using 300 ng of DNA and Lipofectamine 2000 in a 1:2 ratio. Live-imaging analysis was performed 30 h after transfection. Mitotracker green (Invitrogen) was diluted in serum-free medium (OPTI-Mem; Invitrogen) at a final concentration of 200 nM and

incubated for 30 min at 37°C before poly I:C transfection and imaging. Analysis of mRuby-PKR localization during the cell cycle was performed after synchronizing cells in $G_1$ with thymidine (Sigma-Aldrich). Briefly, 5 × 10^4 H4 cells were seeded in a glass-bottom 24-well plate and incubated overnight at 37°C. Cells were pulsed with thymidine (2 mM) for 18 h and chased in complete medium for 9 h at 37°C. A second thymidine pulse was added before live-cell imaging. Cells were washed in phenol-free complete media and images were acquired every 5 min for 16 h. Mitotic events were observed ~8 h after removal of thymidine. 1,6-hexanediol was diluted at a final concentration of 3.5% (vol/vol) in phenol red–free medium and added directly to the cells. IFNβ (R&D Systems) was used at a final concentration of 1,000 units/µl. The AP20187 homodimerizer (Takara) was used at a final concentration of 100 nM for the indicated times. Sodium arsenite (Sigma-Aldrich) was used at a final concentration of 500 µM for 1 h. 4E-T gene silencing was accomplished through transfection of synthetic siRNA. Depletion of 4E-T was performed using a pool of synthetic siRNAs (Dharmacon siGenome-SMART pool; 5′-UUACGAAUCACUGAGGUAGGG-3′ and 5′-UCU CGUGGAUCUACUAUCCTG-3′ and their reverse complements targeting gene NM_019843) transfected with Lipofectamine 2000 following the manufacturer's recommendations. All RNAi experiments were carried out 96 h after transfection.

**MV infection**

The MV C protein knockout (Moraten Vaccine strain MVvac-CKO-GFP; Pfaller et al., 2014) was propagated in Vero cells at an MOI of 0.01. After 48 h, when the cytopathic effect was visible in 100% of the culture, supernatants were collected, clarified by centrifugation at 350 g for 5 min, and filtered through a 0.45-µm surfactant-free cellulose acetate membrane. Aliquots of the viral stock were stored at –80°C. The virus stock titer was determined by fluorescent focus assay on Vero cells. H4 cells expressing

mRuby PKR were infected at an MOI of 0.5 for 48 h and imaged by live-cell imaging confocal microscopy as described below.

## Microscopy
Imaging was performed using an inverted spinning disc confocal microscope (Nikon Ti-Eclipse) equipped with an electron-multiplying charge-coupled device camera (SN:500241; FusionFusion) and environmental control (Okolabs stage top incubator). Live-cell imaging was performed at 37°C and 5% $CO_2$. Images acquisition was performed with a Plan Apochromat 40×, NA 0.95 air objective. For fixed samples, a Plan Apochromat 100×, NA 1.49 oil-immersion objective was used. Live-cell superresolution videos were acquired on a Nikon CSU-W1 SoRa spinning-disk confocal microscope equipped with an electron-multiplying charge-coupled device camera (DU-888; Andor). Images were captured with a Plan Apochromat 60×, NA 1.2 water-immersion objective. All live-cell imaging experiments were performed in phenol-free DMEM complete medium (Gibco).

## FRAP analysis
FRAP analyses were carried out as previously described (Snapp et al., 2003) using a resonant scanning confocal microscope (SP8; Leica) equipped with a Plan Apochromat 60×, NA 1.2 oil-immersion objective. Briefly, $2 \times 10^5$ H4 mRuby PKR cells were grown on a glass-bottom 35-mm dish (MatTek) and transfected with poly I:C as described above 60 min before starting the experiment. Cells were imaged at 37°C and 5% $CO_2$ in phenol-free DMEM complete medium. ROIs for each mRuby-PKR cluster were identified, and two to three clusters per cell were bleached with a 561-nm laser. Because of the high concentration of mRuby-PKR in the clusters, a 15-s continuous bleaching pulse was used to achieve complete bleaching. For normalization, ROIs of similar areas were selected in the cytosol, and the mean intensity of cytosolic ROIs was subtracted from the mean intensity of each bleached cluster. The mobile fraction was calculated as follows:

$$M_f = \frac{I_\infty - I_0}{I_i - I_0},$$

where, $I_\infty$ is the last fluorescence value collected, $I_0$ is the fluorescence value before photobleaching, and $I_i$ is the first value after photobleaching. The immobile fraction was defined as $1 - M_f$.

## Immunofluorescence
$0.8 \times 10^5$ H4 cells were grown on glass coverslips (Thermo Fisher Scientific) and fixed 24 h after plating with 4% paraformaldehyde for 10 min or with ice-cold MeOH for 5 min. Fixed cells were washed with PBS and permeabilized with blocking solution (0.05% saponin, 0.5% BSA, 50 mM $NH_4Cl$, in PBS) for 20 min. Afterward, the samples were incubated 1 h at RT with primary antibodies diluted in blocking solution at the concentrations specified in Table 1. The coverslips were washed with three times with RT room-temperature PBS and incubated with fluorochrome-conjugated secondary antibodies (Alexa Fluor 488, 568, and 647, diluted at 1:500 in blocking solution) and

DAPI (0.1 µg/ml) for 45 min at RT. Cells were washed two times in PBS and one time in $ddH_2O$ before mounting using Mowiol.

## Image quantification and analysis
The proportion of mRuby-PKR in clusters was determined as follows: mRuby-PKR mean fluorescence intensity (MFI) in clusters and total mRuby-PKR MFI per cell were obtained using open-source image processing software Fiji (v2.3) and expressed as a percentage (MFI$^{clusters}$/MFI$^{cell}$ × 100). Colocalization correlation analysis of Edc3 and mRuby-PKR was performed using Fiji (v2.3) as follows: A single-cell ROI was drawn manually for cells with mRuby-PKR clusters, and each cell crop was analyzed individually. After cropping, the image was split into single channels and the Edc3 signal was subtracted from the mRuby-PKR signal using the "image subtraction" plug-in, and the resulting image was used to estimate the diameter of Edc3-free mRuby-PKR clusters using the "analyze particles" plug-in and intermodes-automated thresholding. To quantify the extent of colocalization of mRuby-PKR and Edc3, the signal for mRuby-PKR clusters that are devoid of Edc3 was subtracted from the source image. The diameter of mRuby-PKR– and Edc3-positive clusters was estimated as described above. The number of cells containing mRuby-PKR and FKBP-PKR clusters was estimated using the "multipoint tool" plug-in. After normalizing "cells with clusters" to the "total number of cells" in the field of view, as determined by DAPI staining, we used the Fiji plugin "analyze particles", to count PKR clusters. The area of the FKBP-PKR clusters was calculated using the same plugin. Fluorescent intensity profiles of the indicated ROI were obtained using the plugin "RGB profile plots" in ImageJ. GFP-Dcp1a and mRuby-PKR correlation over time in live-cell images was performed using the Fiji plugin EzColocalization (Stauffer et al., 2018). For each cell, the maximum correlation value over time was selected to plot the data. mRuby-PKR and mRuby-PKR$^{S462A/G466L}$ cluster analysis was performed using Cell Profiler 3.1.8 on at least 30 randomly chosen fields of view for each experimental replicate. Briefly, the analysis pipeline works as follows: (1) Locate nuclei by global thresholding in the Hoechst channel. (2) Identify cells by adaptive Otsu thresholding propagating outwards from the previously identified nuclei. (3) Identify the cytoplasm by subtracting the "cell" signal from the "nucleus" signal. (4) Identify mRuby-PKR clusters by adaptive Otsu thresholding. (5) Measure object size and shape. (6) Assign mRuby-PKR clusters to "parent" cells based on their spatial overlap with the previously identified cytoplasm mask. The data output from Cell Profiler were parsed and analyzed using Python 3.7.

## Western blotting
Cells were washed three times with RT PBS and lysed in Laemmli sample buffer (30 mM Tris-HCl, pH 6.8, 1% SDS, 10% [wt/vol] Glycerol, and bromophenol blue). Lysates were briefly sonicated, 5% 2-mercaptoethanol was added, and the lysates were heated up top 95°C prior to separation by SDS-PAGE. Immunoblotting was performed using nitrocellulose membranes blocked with 1% BSA in TBS-T for 45 min, and incubated at 4°C overnight with the following antibodies: anti-PKR (1:1,000; 3072; Cell Signaling Technology), anti-p-PKR (1:1,000; MA5-32-086;

Table 1.

| Antibody | Manufacturer | Cat. no. | Species | Dilution |
|----------|--------------|----------|---------|----------|
| ATF4 | Cell Signaling Technology | 11815 | Rabbit | 1:400 |
| CHOP | Cell Signaling Technology | 2895 | Mouse | 1:200 |
| Edc3 | Santa Cruz Biotechnology | 271805 | Mouse | 1:600 |
| FLAG | Sigma-Aldrich | F1804 | Mouse | 1:400 |
| FLAG | Abcam | 205606 | Rabbit | 1:400 |
| GM130 | DB Laboratories | 610822 | Mouse | 1:1,000 |
| G3BP1 | Bethyl | A302-033A | Rabbit | 1:1,000 |
| LAMP1 | Cell Signaling Technology | 9091 | Rabbit | 1:1,500 |
| LC3 | Cell Signaling Technology | 2775S | Rabbit | 1:100 |
| PMP70 | Invitrogen | PA1-650 | Rabbit | 1:400 |
| p-eIF2α | Abcam | ab32157 | Rabbit | 1:200 |
| PKR | Santa Cruz Biotechnology | 6282 | Mouse | 1:200 |
| p-PKR | Abcam | ab32036 | Rabbit | 1:200 |
| 4E-T | Bethyl | A300-706A-M-2 | Rabbit | 1:400 |

Sigma-Aldrich or 1:1,000; 32036; Abcam), anti-p-eIF2α (1:1,000; cat. no. 9721; Cell Signaling Technology), anti-eIF2α (1:1,000; cat. no. 9722; Cell Signaling Technology), anti-ATF4 (1:1,000; cat. no. 11815S; Cell Signaling Technology), anti-CHOP (1:1,000; cat. no. 2895S; Cell Signaling Technology), anti-GADD34 (1:1,000; 10449-1-AP; Proteintech), anti-FLAG M2 (1:3,000; F1804; Sigma-Aldrich), anti-puromycin (1:2,000; 2266S; Millipore), anti-β-actin (1:5,000; cat. no. 061M4808; Sigma-Aldrich), anti-GAPDH (1:5,000; cat. No. 8245; Abcam). Membranes were washed three times with TBS-T buffer and incubated at RT for 1 h with horseradish peroxidase–conjugated secondary anti-mouse or anti-rabbit antibodies (1:5,000; Cell Signaling Technology). The membranes were washed three times in TBS-T, and immunoreactive bands were detected by enhanced chemiluminescence.

### Puromycilation of nascent peptides
Puromycilation of nascent peptides was performed as described (Zappa et al., 2019). Briefly, $2 \times 10^5$ FKBP-PKR cells were grown in 6-well plates, and the AP20187 homodimerizer was added 24 h later. 9 μM puromycin was added 1 h after AP20187. Cells were incubated with puromycin for 20 min at 37°C before sample collection. The cells were collected and analyzed as described for Western blotting.

### Immunoprecipitation
$5 \times 10^6$ FKBP-PKR and FKBP-PKR$^{S462A/G466L}$ cells were washed three times in cold PBS and lysed in immunoprecipitation buffer (25 mM Tris-HCl, pH 7.4, 150 mM NaCl, 1 mM EDTA, and 0.5% NP40, supplemented with fresh protease and phosphatase inhibitors). The lysates were clarified for 15 min at 10,000 $g$ at 4°C, and the clarified cell extracts were immunoprecipitated for 3 h at 4°C using FLAG-M2 magnetic beads with end-over-end rotation. The beads were washed six times in immunoprecipitation buffer, and target antigens were recovered by incubating the beads in 100 mM glycine, pH 2.8, for 20 min at 4°C. The

eluates were immediately neutralized with 500 mM Tris, pH 8.0, before separation by SDS-PAGE. Western blot analysis was performed as described above.

### Online supplemental material
Fig. S1 shows generation of a stable cell line expressing fluorescently tagged PKR. Fig. S2 shows analysis of interdependence and colocalization of mRuby-PKR and PBs. Fig. S3 shows colocalization analysis of mRuby-PKR with membrane-bound organelles. Fig. S4 shows that RNA-independent activation of PKR triggers clustering and induces the ISR. Fig. S5 shows that suppression of PKR clustering enhances signaling. Video 1 shows that mRuby-PKR forms clusters in response to poly I:C treatment. Video 2 shows that mRuby-PKR forms clusters during cell division. Video 3 shows that mRuby-PKR clusters and PBs demix in a time-dependent manner. Video 4 shows that mRuby-PKR forms transient associations with mitochondria. Video 5 shows that mRuby-PKR forms transient associations with the ER. Video 6 shows that mRuby-PKR$^{T446A}$ forms clusters in response to poly I:C treatment. Video 7 shows that mRuby-PKR clusters do not recruit mNeon-eIF2α. Video 8 shows that mRuby-PKR clusters recruit mNeon-K3L. Video 9 shows that disruption of PKR's kinase front-to-front (FTF) interfaces suppresses clustering.

## Acknowledgments
We are grateful to Dzwokai Zach Ma and Charles E. Samuel for their kind gift of the MV protein C knockout mutant strain, Carolina Arias for assistance with the MV infections, Julien Bacal for help with data analyses in Cell Profiler, and Meghan Morrisey for providing us with access to their microscopy setups. We also thank Mauro Costa-Mattioli, Adam Frost, Peter Walter, and the members of the Acosta-Alvear and Arias labs for their insightful discussion.

This work is supported by a sponsored research agreement with Calico Life Sciences LLC (D. Acosta-Alvear), a University of California, Santa Barbara Academic Senate Faculty Research grant (D. Acosta-Alvear), departmental start-up funds (D. Acosta-Alvear), and an Otis Williams Postdoctoral Fellowship (F. Zappa).

D. Acosta-Alvear is an inventor on U.S. patent 9708247 held by the Regents of the University of California that describes ISRIB and its analogs. Rights to the invention have been licensed to Calico Life Sciences LLC. For the rest of the authors, there are no competing interests.

Author contributions: D. Acosta-Alvear supervised the research. F. Zappa and D. Acosta-Alvear designed the experiments, analyzed the data, and wrote the manuscript. N.L. Muniozguren generated stable cell lines. M.Z. Wilson provided access to instrumentation. M.S. Costello and J.C. Ponce-Rojas produced the constructs used in the study.

Submitted: 21 November 2021

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

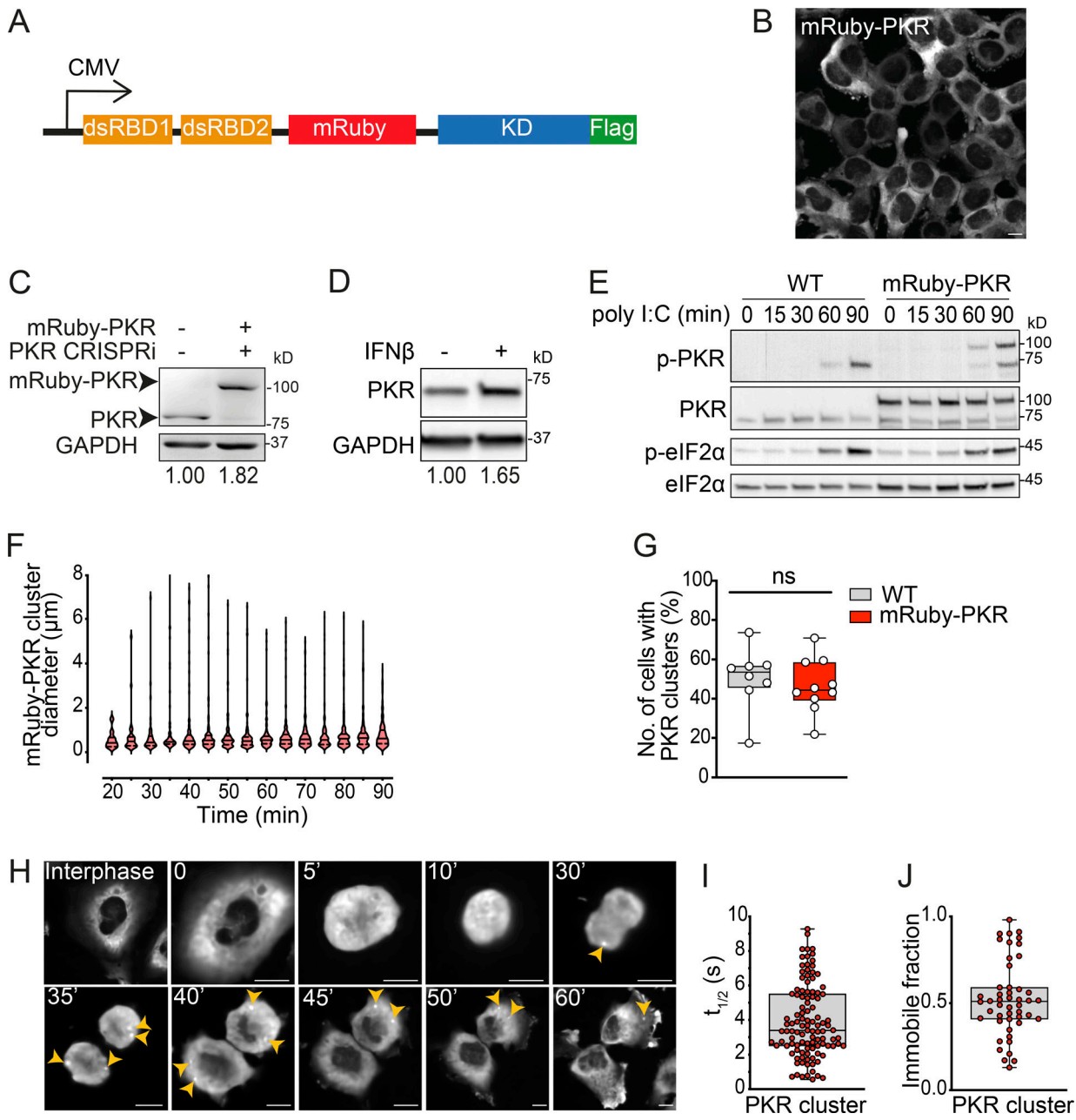

Figure S1.   **Generation of a stable cell line expressing fluorescently tagged PKR. (A)** Schematic representation of the expression construct encoding mRuby- and FLAG-tagged PKR of human origin. The mRuby fluorescent protein (236 aa) was inserted between residues 221 and 222 of human PKR. **(B)** Representative micrograph showing that mRuby-PKR is a cytosolic soluble protein. Scale bar: 10 μm. **(C)** Western blot showing the level of expression of mRuby-PKR compared with that of endogenous PKR. The relative protein amount determined by densitometry is shown below the blots. Endogenous PKR was depleted using CRISPRi. GAPDH, loading control. **(D)** Western blot analysis showing endogenous PKR induction in wild-type H4 cells treated with IFNβ for 16 h. GAPDH, loading control. Metrics as in C. **(E)** Western blot analysis comparing the activity and kinetics of endogenous PKR and mRuby-PKR in H4 cells treated with poly I:C. **(F)** Violin plots showing the diameter of mRuby-PKR clusters in H4 cells stably expressing mRuby-PKR and treated with poly I:C. **(G)** Quantification of the number of cells with endogenous PKR and mRuby-PKR clusters upon 90 min of poly I:C treatment ($N$ = 3 experiments, $n$ > 800; unpaired Student's $t$ test, nonparametric). **(H)** Representative time-lapse micrographs showing the formation of mRuby-PKR clusters (yellow arrowheads) in dividing H4 mRuby-PKR cells synchronized with thymidine. Scale bar: 10 μm. **(I)** Quantification of the half-life of mRuby-PKR in clusters after photobleaching. **(J)** Quantification of the mRuby-PKR immobile fraction in clusters after photobleaching. For H and I, $N$ = 3 experiments, $n$ = 30. Source data are available for this figure: SourceData FS1.

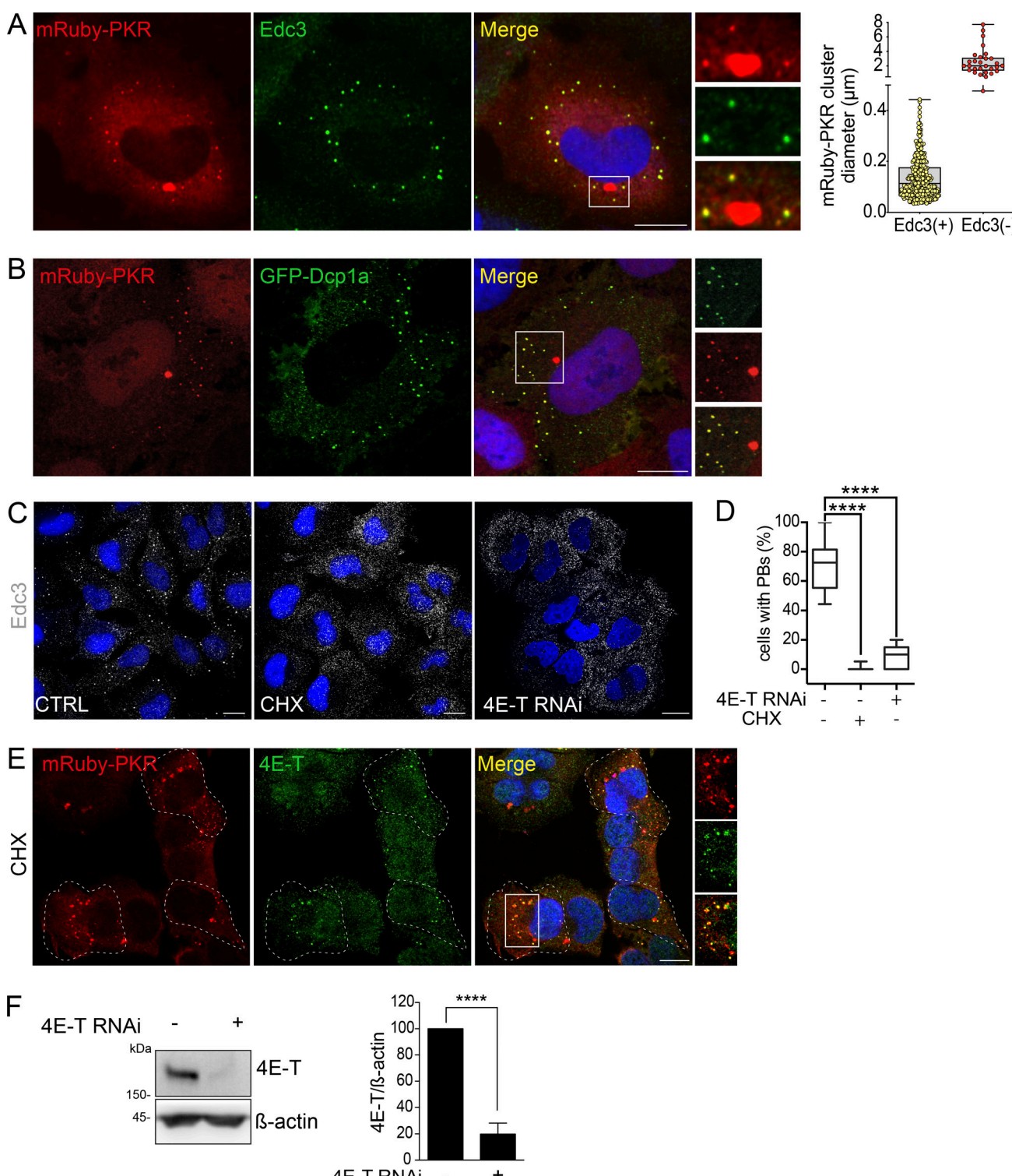

Figure S2. **Analysis of interdependence and colocalization of mRuby-PKR and PBs. (A)** Representative immunofluorescence images showing two populations of mRuby-PKR clusters based on their diameter and association with Edc3. The image crop shows a close-up of these two populations. Scale bar: 10 µm. The right panel shows the quantification of the data (*n* = 30 cells). **(B)** Representative micrographs of fixed cells coexpressing mRuby-PKR and GFP-Dcp1a. The image crop shows two mRuby-PKR cluster populations, those that associate with GFP-Dcp1a and those that do not. Scale bar: 10 µm. **(C)** Representative immunofluorescence images showing pharmacological (CHX) or genetic (4E-T RNAi) knockdown of PBs assessed by Edc3 staining. Scale bar: 10 µm. **(D)** Quantification of the data in C. (*N* = 3 experiments, *n* > 500; ****, P < 0.0001, unpaired Student's *t* test, nonparametric). **(E)** Representative micrographs showing that mRuby-PKR clusters recruit 4E-T after pharmacologic depletion (CHX) of PBs. Scale bar: 10 µm. **(F)** Western blot showing the extent of knockdown of 4E-T KD by RNAi. Right: Quantification of the data (mean and SEM, *N* = 3 experiments; ****, P < 0.0001, unpaired Student's *t* test, nonparametric). Source data are available for this figure: SourceData FS2.

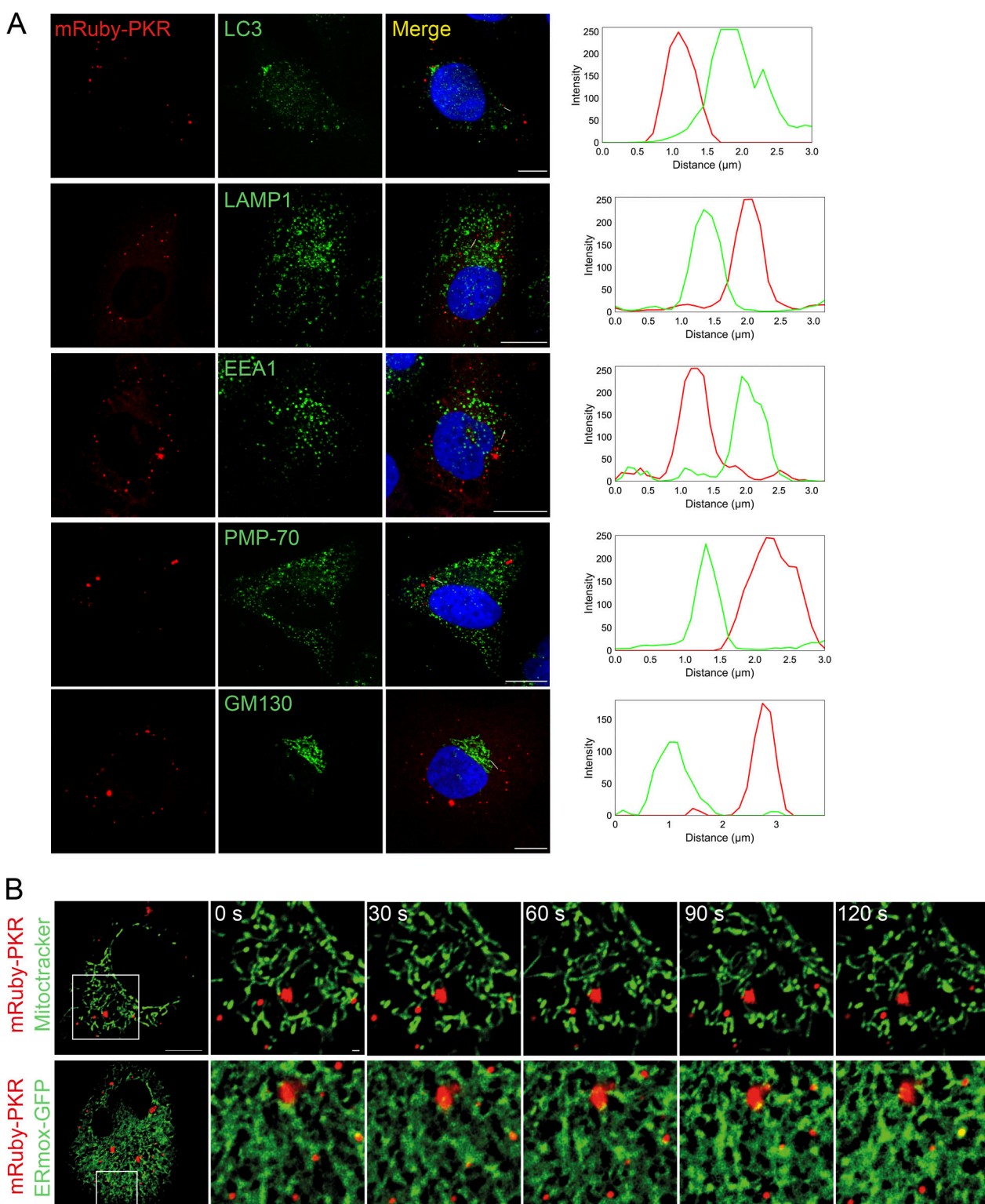

Figure S3. **Colocalization analysis of mRuby-PKR with membrane-bound organelles. (A)** Representative immunofluorescence images showing that mRuby-PKR clusters do not colocalize with the autophagosomes (LC3), lysosomes (LAMP1), early endosomes (EEA1), peroxisomes (PMP-70), or cis-medial Golgi apparatus (GM130). The plots of signal intensity of mRuby-PKR clusters (red) and organelle markers (green) as a function of distance were prepared as in Fig. 2 A. The ROIs used for metrics are indicated with a white line. Scale bar: 10 μm. **(B)** Representative time-lapse micrographs showing transient association of mRuby-PKR clusters with mitochondria (Mitotracker) or the ER (ERmox-GFP). Scale bar: 10 μm; inset: 2 μm.

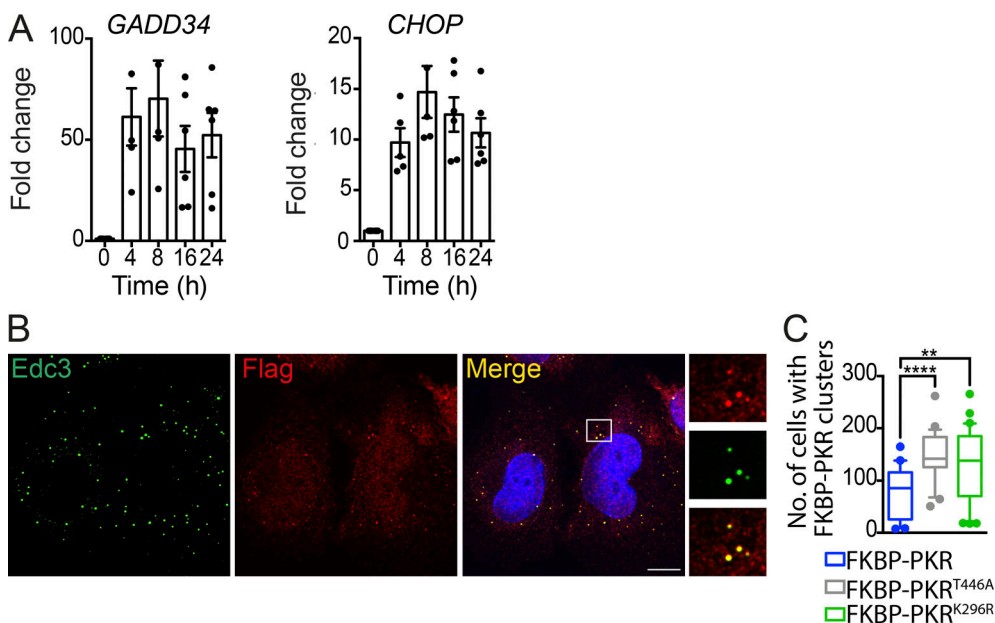

Figure S4. **RNA-independent activation of PKR triggers clustering and induces ISR. (A)** Quantitative real-time PCR analysis of GADD34 and CHOP levels after forced-dimerization of FKBP-PKR (mean and SEM, $N = 5$ experiments). **(B)** Representative immunofluorescence images showing colocalization of FLAG-tagged FKBP-PKR and Edc3. Image crop on right: close-up. Scale bar: 10 μm. **(C)** Quantification of the number of catalytically dead FKBP-PKR clusters 10 min after dimerizer addition ($N = 3$ experiments, $n > 1,000$; **, $P < 0.01$; ****, $P < 0.0001$, unpaired Student's $t$ test, nonparametric).

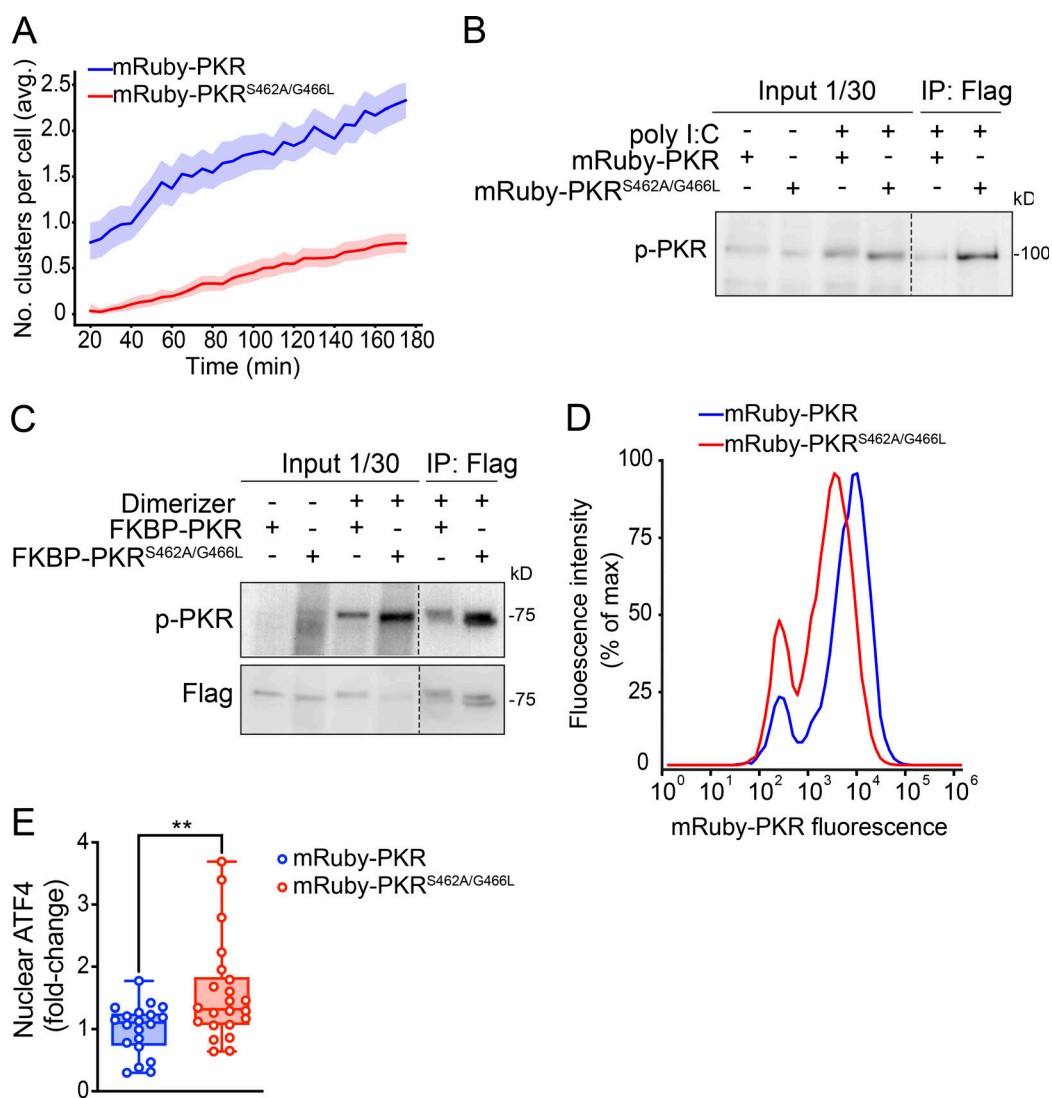

Figure S5. **Suppression of PKR clustering enhances signaling. (A)** Image quantification showing that the mutations in PKR's front-to-front (FTF) kinase interfaces significantly reduce the number of PKR clusters in cells. The data were binned and are shown as the mean and 95% confidence interval bands ($n$ = 2,000 cells). **(B)** Immunoprecipitation (IP) analysis of the extent of phosphorylation of mRuby-PKR and FTF mutant mRuby-PKR upon 90-min poly I:C treatment. **(C)** Same as B for FKBP-PKR and FTF mutant FKBP-PKR upon 60 min of dimerizer treatment. **(D)** Flow cytometry histograms showing the relative fluorescent intensity of cells expressing wild-type mRuby-PKR (blue trace) and FTF mutant mRuby-PKR (red trace). **(E)** Quantification of nuclear ATF4 signal in immunofluorescence analyses carried out in H4 cells expressing mRuby-PKR or FTF mutant mRuby-PKR treated with poly I:C. ($N$ = 3 experiments, $n$ > 600; **, $P$ < 0.01, unpaired Student's $t$ test, nonparametric). Source data are available for this figure: SourceData FS5.

Video 1. **mRuby-PKR forms clusters in response to poly I:C treatment.** Time-lapse confocal fluorescence microscopy of H4 neuroglioma cells stably expressing mRuby-PKR showing the formation of mRuby-PKR clusters upon transfection with 2 µg/ml poly I:C. The squares indicate areas of cluster merging (bottom right) and segregation (upper left). Images were acquired every minute for 105 min. Playback, 5 fps. Scale bar: 10 µm. Relates to Fig. 1 A.

Video 2. **mRuby-PKR forms clusters during cell division.** Time-lapse confocal fluorescence microscopy of H4 neuroglioma cells stably expressing mRuby-PKR showing formation of mRuby-PKR during cell division. Cells were synchronized using the double-thymidine protocol. Images were acquired every 5 min for 135 min. Playback, 3 fps. Scale bar: 10 µm. Relates to Fig. 1 H.

Video 3. **mRuby-PKR clusters and PBs de-mix in a time-dependent manner.** Time-lapse confocal fluorescence microscopy of H4 neuroglioma cells stably expressing mRuby-PKR showing that mRuby-PKR clusters (in red) and PBs (GFP-Dcp1a, in green) de-mix in a time-dependent manner. Images were acquired every minute for 90 min. Playback, 5 fps. Scale bar: 10 µm. Relates to Fig. 2 B.

Video 4. **mRuby-PKR forms transient associations with mitochondria.** Time-lapse superresolution fluorescence microscopy of H4 mRuby-PKR cells transfected with 2 µg/ml poly I:C 60 min before starting the imaging. The video shows that mRuby-PKR clusters transiently associate with mitochondria (Mitotracker green). Images were acquired every second for 5 min. Playback, 5 fps. Scale bar: 2 µm. Relates to Fig. S3 B.

Video 5. **mRuby-PKR forms transient associations with the ER.** Time-lapse superresolution fluorescence microscopy of H4 mRuby-PKR cells transiently expressing ERmox-GFP and transfected with 2 µg/ml poly I:C 60 min before starting the imaging. The video shows that mRuby-PKR clusters transiently associate with the ER. Images were acquired every 5 s for 2 min. Playback, 5 fps. Scale bar: 2 µm. Relates to Fig. S3 B.

Video 6. **mRuby-PKRT446A forms clusters in response to poly I:C treatment.** Time-lapse confocal fluorescence microscopy of H4 neuroglioma cells stably expressing mRuby-PKRT446A showing the formation of mRuby-PKRT446A clusters upon transfection with 2 µg/ml poly I:C. Images were acquired every 5 min for 90 min. Playback, 5 fps. Scale bar: 10 µm. Relates to Fig. S3 C.

Video 7. **mRuby-PKR clusters do not recruit mNeon-eIF2α.** Time-lapse confocal fluorescence microscopy of H4 cells stably expressing mRuby-PKR (red) and mNeon-eIF2α (green) and transfected with 2 µg/ml poly I:C. The video shows that mNeon-eIF2α is not enriched in mRuby-PKR clusters. Images were acquired every 5 min for 90 min. Playback, 5 fps. Scale bar: 10 µm. Relates to Fig. 4 A.

Video 8. **mRuby-PKR clusters recruit mNeon-K3L.** Time-lapse confocal fluorescence microscopy of H4 cells stably expressing mRuby-PKR (red) and mNeon-K3L (green), and transfected with 2 µg/ml poly I:C. The video shows that mNeon-K3L is recruited into mRuby-PKR clusters with a lag time of ~10 min after their formation. Images were acquired every 5 min for 90 min. Playback, 5 fps. Scale bar: 10 µm. Relates to Fig. 4 D.

Video 9. **Disruption of PKR's kinase front-to-front (FTF) interfaces suppresses clustering.** Time-lapse confocal fluorescence microscopy of H4 cells stably expressing mRuby-PKR (left) and mRuby-PKRS462A/G466L (right), showing dramatically reduced ability of mRuby-PKRS462A/G466L to cluster upon transfection with 2 µg/ml poly I:C. Images were acquired every 5 min for 90 min. Playback, 5 fps. Scale bar: 10 µm. Relates to Fig. 5 B.

