## [Peer Review File · The Journal of Cell Biology]

Signaling by the integrated stress response kinase PKR is fine-tuned by dynamic clustering

Francesca Zappa, Nerea L. Muniozguren, Maxwell Wilson, Michael Costello, Jose Ponce-Rojas, and Diego Acosta-Alvear

Corresponding Author(s): Diego Acosta-Alvear, University of California Santa Barbara; Diego Acosta-Alvear, University of California Santa Barbara; and Francesca Zappa, University of California, Santa Barbara

Review Timeline:

Submission Date:	2021-11-21
Editorial Decision:	2022-01-05
Revision Received:	2022-02-15
Editorial Decision:	2022-03-25
Revision Received:	2022-04-01

Monitoring Editor: Judith Frydman

Scientific Editor: Tim Fessenden

Transaction Report:

DOI: <https://doi.org/10.1083/jcb.202111100>

January 5, 2022

Re: JCB manuscript #202111100

Dr. Diego Acosta-Alvear
University of California Santa Barbara
Molecular, Cellular and Developmental Biology
2115 Life Sciences Building
Santa Barbara 93106

Dear Dr. Acosta-Alvear,

Thank you for submitting your manuscript entitled "Signaling by the integrated stress response kinase PKR is fine-tuned by dynamic clustering". The manuscript was assessed by expert reviewers, whose comments are appended to this letter. We invite you to submit a revision if you can address the reviewers' key concerns, as outlined here.

You will see that all reviewers acknowledge the conceptual advance conveyed by this work. Both Rev3 and Rev1 seek clarification on the timing of PKR cluster formation, which is proposed to stop/attenuate the integrated stress response vs eIF2a phosphorylation. This point is central to the advance proposed by the paper and should be addressed in revisions. Next, point 3 by Rev3 requests an important control to eliminate artifacts from mRuby tagging. Given that Rev1's points are also encompassed by Rev3, the first 3 major points by Rev3 should guide the revisions. Point 4 by Rev3 seeks clarification of the minimal signal needed to drive PKR clustering, which Rev3 suggests addressing by use of an inactive mutant PKR. This would further clarify the point that dsRNA is sufficient to drive clustering. While this would add some clarity to that section of the paper, it is an option for you to address should you wish. All minor points are left to your discretion but were generally constructive and would likely improve the work.

GENERAL GUIDELINES:

Text limits: Character count for an Article is < 40,000, not including spaces. Count includes title page, abstract, introduction, results, discussion, acknowledgments, and figure legends. Count does not include materials and methods, references, tables, or supplemental legends.

Figures: Articles may have up to 10 main text figures. Figures must be prepared according to the policies outlined in our Instructions to Authors, under Data Presentation, <https://jcb.rupress.org/site/misc/ifora.xhtml>. All figures in accepted manuscripts will be screened prior to publication.

*****IMPORTANT:** It is JCB policy that if requested, original data images must be made available. Failure to provide original images upon request will result in unavoidable delays in publication. Please ensure that you have access to all original microscopy and blot data images before submitting your revision. ***

Supplemental information: There are strict limits on the allowable amount of supplemental data. Articles may have up to 5 supplemental figures. Up to 10 supplemental videos or flash animations are allowed. A summary of all supplemental material should appear at the end of the Materials and methods section.

Please note that JCB now requires authors to submit Source Data used to generate figures containing gels and Western blots with all revised manuscripts. This Source Data consists of fully uncropped and unprocessed images for each gel/blot displayed in the main and supplemental figures. Since your paper includes cropped gel and/or blot images, please be sure to provide one Source Data file for each figure that contains gels and/or blots along with your revised manuscript files. File names for Source Data figures should be alphanumeric without any spaces or special characters (i.e., SourceDataF#, where F# refers to the associated main figure number or SourceDataFS# for those associated with Supplementary figures). The lanes of the gels/blots should be labeled as they are in the associated figure, the place where cropping was applied should be marked (with a box), and molecular weight/size standards should be labeled wherever possible.

As you may know, the typical timeframe for revisions is three to four months. However, we at JCB realize that the implementation of social distancing and shelter in place measures that limit spread of COVID-19 also pose challenges to scientific researchers. Lab closures especially are preventing scientists from conducting experiments to further their research. Therefore, JCB has waived the revision time limit. We recommend that you reach out to the editors once your lab has reopened to decide on an appropriate time frame for resubmission. Please note that papers are generally considered through only one revision cycle, so any revised manuscript will likely be either accepted or rejected.

Thank you for this interesting contribution to Journal of Cell Biology. You can contact us at the journal office with any questions, cellbio@rockefeller.edu or call (212) 327-8588.

Sincerely,

Judith Frydman
Monitoring Editor
Journal of Cell Biology

Tim Fessenden
Scientific Editor
Journal of Cell Biology

Reviewer #1 (Comments to the Authors (Required)):

This manuscript by Zappa and colleagues addresses the role of PKR's higher-order structures during ISR activation. The authors take advantage of the recent observation that upon ligand binding PKR forms cluster structures, which are mediated by front-to-front interactions (unlike the typical front-to-back ones required for dimer formation during ISR activation). Using a series of imaging analyses, they show that poly I:C and mock-viral infections result in the formation of PKR oligomeric structures that appear to share some of the processing bodies factors, but not stress granules ones. The authors then went on to claim that disruption of cluster formation results in increased eIF2 phosphorylation.

Overall, the paper was well written, and the data appear to be of high quality. However, I do not think the authors provided enough support for their model that cluster formation is used to temper ISR activation and furthermore they did not provide any data on how this might be regulated. I should emphasize that the model the authors propose (if correct) is novel, but their data are largely descriptive and can be interpreted by multiple models.

Specific comments:

- 1- Arguably, the authors' model is largely based on one piece of data showing that mutations that disrupt oligomer formation lead to increased eIF2 phosphorylation. How do the authors exclude the fact that these mutations can affect other functions of PKR? Indeed, Mayo et al. (2019), whose report was the basis for this paper, noted that the effect of these mutations on dimer formation could not be assessed. Furthermore, the increase in phosphorylation is modest.
- 2- Can the authors elaborate on what they mean by "these results support a model in which PKR clustering buffers downstream signaling, which may enable proofreading the ISR"? How does proofreading work? Do clusters recruit phosphatase to dephosphorylate eIF2? How are they regulated? As far as I can tell, there were no data supporting a proofreading mechanism, which would be very novel.
- 3- Related to the above point, a major claim by the authors is that cluster formation dampens ISR, but their formation is induced by the very same signal that activates ISR. The authors need to clarify how cluster formation is regulated.
- 4- Also related to their model. This statement in the discussion "possible explanation for this observation is that PKR's phosphatases can be recruited to the clusters to suppress excessive PKR signaling" is inconsistent with the data in Figure 1D, showing p-PKR to localize with PKR clusters with no apparent decrease in intensity relative to diffuse p-PKR.
- 5- Overall, the manuscript provides compelling evidence that PKR forms higher-order structures that share features with PB. These observations are not new, and they have been documented by others.

Minor:

1- The authors should attempt to reconcile their observation "disruption of in vivo PKR clustering did not suppress PKR's self-phosphorylation but rather enhanced it" with earlier reports showing the opposite in vitro.

2- Page 8 typo, there is an extra "corroborated" after poly I:C

Reviewer #2 (Comments to the Authors (Required)):

In this paper, Zappa et al. present data in support of a new and exciting feature of PKR signaling that may enable the control of PKR-eIF2 interactions to fine-tune ISR signaling. The authors show that PKR forms dynamic assemblies in response to dsRNA and that these assemblies, which the authors refer to as "PKR clusters", contain phosphorylated (i.e., active) PKR. Moreover, the PKR assemblies can recruit P-body components, but they do not require P-bodies, indicating that "clustering" is an intrinsic property of active PKR. Most surprising is the finding that eIF2, a canonical PKR substrate, is not present in the clusters and that impeding PKR cluster assembly through mutagenesis enhances eIF2 phosphorylation. The results support a new model in which PKR clustering may provide a mechanism to regulate PKR (and ISR signaling) by adjusting PKR-eIF2 encounters. The data are of excellent quality, and the findings significantly increase our knowledge of the mechanisms by which PKR regulate downstream signaling. I consider this manuscript to be appropriate for the readership of JCB, and I recommend its publication. That said, I have a few minor points that need to be addressed before publication.

Minor points:

1. One could question whether endogenous PKR would exhibit the same behavior as the ruby-tagged PKR or whether clustering is a product of a roughly 2-fold overexpression. However, interferons induce PKR during the natural physiological response to viral infection, which support the main conclusion and is likely to reflect the natural behavior of the protein. The authors could add a sentence in the discussion regarding this issue.
2. The authors show that disruption of PKR clusters enhances eIF2 phosphorylation. It will strengthen the paper if they show the enhancement of the canonical ISR downstream signaling. e.g., increased levels of ATF4.
3. The authors demonstrate that PKR clusters can form in the absence of P-bodies, but can P-bodies form in the absence of PKR?
4. The data with the measles mutant virus convincingly show that PKR forms clusters in response to natural dsRNAs. The data of PKR clustering in mitotic cells supports the point that endogenous inputs, probably natural nuclear dsRNAs, also drive "PKR clustering". Both data sets substantiate the notion that PKR forms clusters in response to naturally occurring dsRNAs, but I find the virus infection data more compelling. The authors should consider moving the cell division data to the supplement.

Reviewer #3 (Comments to the Authors (Required)):

This is an interesting and informative study that demonstrates for the first time that PKR assembles into cytosolic clusters upon stimulation by dsRNA. These clusters are dynamic and distinct from processing bodies. Formation of these clusters is inhibited by mutations designed to interrupt a front-to-front kinase domain interaction, suggesting that this interaction is involved in cluster assembly. A model is presented where PKR clusters function to sequester PKR in a pool incapable of phosphorylating its primary substrate, eIF2 α and thus control downstream signaling.

Overall, the work is solid and the major conclusions are supported by high quality data. However, as noted below, there are some problems with the model for the role of the clusters in attenuating substrate phosphorylation and some additional experiments are required to round out the study.

Major Issues:

- 1) The model where PKR clustering attenuates eIF2 α phosphorylation has issues. It is primarily based on the observation that mNeon-eIF2 α is excluded from the PKR clusters. These results are interpreted to indicate that the clusters act as "enzyme sinks" that regulate the extent of eIF2 α phosphorylation and control the timing and amplitude of PKR signaling. This model begs the question of how eIF2 α can become phosphorylated if it cannot access the active, and presumably autophosphorylated, enzyme that is sequestered in the clusters. Is eIF2 α phosphorylation mediated by active monomeric or dimeric PKR prior to incorporation into clusters? If so, then eIF2 α phosphorylation should precede cluster formation; however, the kinetics of eIF2 α phosphorylation appear quite slow after addition of poly(I:C), with a lag-time of about 60 minutes (Fig. 5c). In contrast, cluster formation is rapid (Fig. 5B, S5A). Thus, this model is not consistent with the data.
- 2) It is claimed that the S462A/G466L double mutant of PKR is active, and in fact it appears more active than wild type at phosphorylating eIF2 α (Figs. 5C, 5D). However, the in vitro studies (Mayo et al., 2019) indicate that S462A PKR is partially active and G466L is essentially inactive with respect to autophosphorylation. It is possible that this discrepancy arises due to the activity of or more of the three other eIF2 α kinases. Thus, the authors need to directly demonstrate the effect of the double mutation on the initial step in PKR activation, autophosphorylation. This is a crucial piece of data that is missing from this study.

Was this experiment omitted because of the data in Figure S5B indicating that the expression level of the double mutant of mRuby-PKR is several fold lower than the wild-type?

3) The study is predicated on the assumption that the mRuby-PKR construct accurately reflects the behavior of native PKR. The insertion of a folded mRuby domain into the unstructured linker lying between dsRBD2 and the kinase domain may affect its dsRNA binding, self-association, and enzymatic activity. Figure S1B shows that the expression levels of endogenous PKR and mRuby-PKR are close (within a factor of 2). One way to assess mRuby-PKR activity relative to the native enzyme is to compare the extent of autophosphorylation and eIF2 α phosphorylation of cells expressing either form upon stimulation with poly(rI:rC).

4) The present work cannot distinguish whether PKR cluster formation requires autophosphorylation or can be driven solely by assembly on dsRNA. A simple way to assess this is to compare cluster formation in cells expressing either wild-type mRuby-PKR or an inactive mutant of mRuby-PKR (e.g., K296R). This experiment would greatly enhance our understanding of the clustering process.

Minor Issues:

1) page 4: The statement describing PKR clusters as "dynamic coalescences that are reminiscent of coacervates" is vague. Do the authors believe that the clusters represent liquid-liquid phase separated coacervates? Alberti et al. (2019) point out several caveats regarding the use of 1,6 hexanediol and FRAP as criteria for a liquid-liquid phase separation.

2) Figure 1D: It is interesting that PKR phosphorylation appears to occur only in the clusters and the bulk of the mRuby-PKR in the cytoplasm does not appear to be phosphorylated. This appears to support a model where cluster formation requires autophosphorylation and not just assembly on dsRNA (See major issue #4).

3) Figure 1F: The Y-axis labels on the right side have a typo (change 1.0 to 10.0). How were these data analyzed to determine that 45% of the cells formed clusters?

4) Figure 1G: The cluster formation in cells undergoing mitosis is not very convincing. The intensities of the clusters are barely above the background fluorescence (soluble monomeric mRuby-PKR?). In any case, the statement in the discussion (page 9) that this data indicates that cluster formation is driven by endogenous dsRNAs is speculative. Factors other than the release of nuclear dsRNAs could drive cluster formation during cell division.

5) Figure 1H: The Y-axis labels at the breakpoint are incorrect. Also, the maximum for a normalized intensity should be 1, not 100.

6) Page 4: The rate of exchange between clusters and cytosolic pools of mRuby-PKR was measured from FRAP and is stated as 3.93 s. First, this is a half-life, not a rate. Second, the distribution of the half-lives (Fig. S1E) is so broad that an average half-life is not meaningful.

7) Figure S1: In the caption "F" is labeled as "E".

8) Figure S1B: The abbreviation PKR KD is meant to refer to knockdown but can also be interpreted as kinase domain.

9) Page 3: It is stated that the size ranges from 0.22 to 5 μ m in diameter but Fig. S1D indicates that the size range is very large and goes up to \sim 8 μ m.

10) Page 5: Reference to S2A and S2B in the text should be S2.1A and S2.1B.

11) Page 5: It is stated that ejection of mRuby-PKR from GFP-Dcp1a-containing clusters occurs in about 15 minutes yet the images in Fig. 2B only begin at 25 minutes.

12) Page 5: The origin of the discrepancy between the extent of colocalization of PB and PKR clusters determined by fixed and live cell imaging should be clarified.

13) Page 6: The extent of depletion of PBs by CHX and knockdown of 4E-T is not clear in Fig. S2.1C. Some quantitation is warranted.

14) Figure 2D: How can the violin plots exceed the value of 100 when the scale is % maximum?

15) Page 6: The reference to Lee et al, 2020 should be 2020b.

16) Page 7, line 2: Replace RBDs with dsRBDs.

17) Figure 4B: It appears that the phospho-eIF2 α staining is prominent in the nucleus in the steady-state image. This protein should be primarily cytosolic.

18) Figure 5A: The effects of the mutations on higher-order assembly of PKR are not well represented by the cartoon. The WT representation implies that the interaction involves the phosphorylated residue(s) in the activation loop. However, the crystal structures exhibiting a front-to-front interface were obtained for the unphosphorylated PKR kinase so this interaction cannot require autophosphorylation. The significance of the two different arrows on the right (mutant) side of the figure is unclear. Does it indicate that there is still some sort of front-to-front interaction in the presence of the two mutations? The data in Figure 5B argues that the interaction is pretty well gone.

19) Figure S5B: The data for wild-type mRuby-PKR cluster formation does not agree with Fig. 1B. The number of clusters per cell is much less and the kinetics are much slower. Why?

20) Figure 5E: It is hard to see by eye that there is any difference in the staining of the cells expressing WT and mutant FKBP-PKR. At the resolution and magnification of the images both seem to show clusters. Thus, the dramatic effect in the quantitation shown in Figure 5F is suspect.

21) Figure 5: There are two part D and part E descriptions in the legend.

22) Figure S5C is not described fully and there are multiple PKR bands in the p-PKR immunoblot. It does not clearly demonstrate enhanced autophosphorylation of the double mutant of FKBP-PKR. If it did show this, it would contradict Figure 5G.

23) The reference for Belyy et al. (2021) is incorrect.

24) There is no Figure S4.

We thank the editors and reviewers for the detailed assessment of our work. We address each Reviewer's comments, point-by-point below.

Reviewer #1 (Comments to the Authors (Required)):

This manuscript by Zappa and colleagues addresses the role of PKR's higher-order structures during ISR activation. The authors take advantage of the recent observation that upon ligand binding PKR forms cluster structures, which are mediated by front-to-front interactions (unlike the typical front-to-back ones required for dimer formation during ISR activation). Using a series of imaging analyses, they show that poly I:C and mock-viral infections result in the formation of PKR oligomeric structures that appear to share some of the processing bodies factors, but not stress granules ones. The authors then went on to claim that disruption of cluster formation results in increased eIF2 α phosphorylation.

Overall, the paper was well written, and the data appear to be of high quality. However, I do not think the authors provided enough support for their model that cluster formation is used to temper ISR activation and furthermore they did not provide any data on how this might be regulated. I should emphasize that the model the authors propose (if correct) is novel, but their data are largely descriptive and can be interpreted by multiple models.

We thank the Reviewer for the insights and we have addressed the comments below.

Specific comments:

1- Arguably, the authors' model is largely based on one piece of data showing that mutations that disrupt oligomer formation lead to increased eIF2 α phosphorylation. How do the authors exclude the fact that these mutations can affect other functions of PKR?

We cannot exclude that the mutations we engineered into PKR might affect other PKR functions, for example activation of other signaling pathways such as NF κ B, which we have not investigated in this work because they lie outside of its scope. Nevertheless, we can reliably conclude that our analyses of the best characterized signaling pathway downstream of PKR, initiated by phosphorylation of eIF2 α , are reflective of the novel aspects of PKR signaling we describe in our work. Our analyses of the eIF2 α phosphorylation status performed in cells expressing mRuby-PKR^{S462A/G466L} show a modest but significant increase in P-eIF2 α and ATF4 levels (Figs. 5C, D; S5E), and are consistent with our data in Figs. 5G, H showing that FKBP-PKR^{S462A/G466L} lead to enhanced P-eIF2 α levels upon ligand administration. The essentially identical results obtained with different engineered proteins harboring the same mutations argue in favor of our conclusion that preventing PKR clustering accelerates and enhances—albeit modestly—eIF2 α phosphorylation.

Indeed, Mayo et al. (2019), whose report was the basis for this paper, noted that the effect of these mutations on dimer formation could not be assessed. Furthermore, the increase in phosphorylation is modest.

PKR dynamics can offer an explanation for the modest increase in eIF2 α phosphorylation we observed upon PKR cluster inhibition. Our FRAP data indicates (i) a highly dynamic diffusible PKR pool in clusters, with half-life of 3.92 seconds (Figs. 1I; S1H), (ii) we calculated that 5% of total PKR is recruited into clusters (Fig. 1A), and (iii) eIF2 α is excluded from PKR clusters (Figs. 4A, B; Video 7). One interpretation of these results is that active PKR dimers freely exchange with the cluster, and those dimers that

“escape” the cluster are capable of interacting with eIF2 α leading to its phosphorylation, as illustrated in our model figure. Considering that the substrate-inaccessible fraction of active PKR is 5% of total PKR, then the modest increase in P-eIF2 α we reported is consistent with our data. Two recent publications on the behavior of IRE1, a stress sensor kinase/RNase that clusters upon activation, lend support to our findings. First, the lab of Peter Walter (UCSF-HHMI) estimates that 5% of IRE1 is recruited into clusters, leaving the rest of the enzyme pool free to interact with its substrate, the mRNA encoding the transcription factor XBP1 (Belyy et al., *PNAS* 2020 PMID: 31871156). Second, Tomás Aragón, Jeff Chao, and Franka Voigt show that IRE1 does not meet XBP1 mRNA in IRE1 clusters (Gómez-Puerta et al., *bioRxiv* 2021; DOI: <https://doi.org/10.1101/2021.11.15.468613>). Therefore, it is plausible that stress sensor clustering provides a mechanism to regulate enzyme-substrate encounters to fine-tune downstream signaling, as we discuss in our manuscript.

2- Can the authors elaborate on what they mean by "these results support a model in which PKR clustering buffers downstream signaling, which may enable proofreading the ISR"? How does proofreading work? Do clusters recruit phosphatase to dephosphorylate eIF2 α ? How are they regulated? As far as I can tell, there were no data supporting a proofreading mechanism, which would be very novel.

We have clarified our interpretation in the manuscript abstract, in lines 26-28, where we state “Together, these results support a model in which PKR clustering limits encounters between PKR and eIF2 α to buffer downstream signaling and prevent the ISR from misfiring.”

3- Related to the above point, a major claim by the authors is that cluster formation dampens ISR, but their formation is induced by the very same signal that activates ISR. The authors need to clarify how cluster formation is regulated.

We have indicated that PKR cluster formation is driven by ligand binding. Ligand binding to PKR is the first step in the activation of a natural (dsRNA-driven) or “virtual” (FKBP-ligand-driven) ISR. A recent pre-print from the laboratory of Roy Parker submitted after our work (Corbet et al. *Biorxiv* 2022; <https://doi.org/10.1101/2022.01.14.476399>) is in concordance with our data that ligand binding (i.e., dsRNA binding) drives the formation of PKR clusters, even in kinase activity deficient PKR mutants (Video 6 and Figs. 3H, S3C on the new version of our manuscript; Figs. 4B, G, H in Corbet et al. 2022). We also measured PKR cluster formation and ISR signaling in poly I:C dose-response experiments. These experiments showed a direct correlation between ligand concentration and the formation of PKR clusters. However, the levels of nuclear ATF4 reached saturation at a relatively low poly I:C concentration of 0.5 μ g/ml, indicating that the ATF4-dependent transcriptional response uncouples from PKR cluster formation as the number of PKR clusters increases (figure below). A possible interpretation of these results is that cluster formation is not required to initiate PKR signaling and clustering is a consequence of sustained PKR activation that might be preventing ISR hyperactivation.

Quantification of number of cells with PKR clusters express as percent of the total population (red trace) and ATF4 nuclear signal intensity (green trace) in cells treated with poly I:C (mean and 95% CI bands; n>500).

4- Also related to their model. This statement in the discussion "possible explanation for this observation is that PKR's phosphatases can be recruited to the clusters to suppress excessive PKR signaling" is inconsistent with the data in Figure 1D, showing p-PKR to localize with PKR clusters with no apparent decrease in intensity relative to diffuse p-PKR.

The high concentration of PKR molecules in a cluster boosts the mean intensity of P-PKR signal in them but does not exclude the presence of P-PKR in the cytosol (see our explanation above about the amount of PKR localized to the clusters). Our FRAP analyses in Figure 1I and S1I show a high exchange rate of PKR molecules with the cytosol, suggesting that active PKR dimers in the cluster are in dynamic equilibrium with the cytosolic pool of active PKR dimers or inactive (i.e., non-phosphorylated) PKR monomers. Such continuous exchange does not rule out the possibility of coincident detection of P-PKR and its phosphatase in the cluster. Phosphatase-substrate encounters have been reported in membrane-less organelles, as occurs with PP1 and its substrate Gemin8 in the Cajal bodies (Renois  et al., *JCS* 2012; PMID: 22454514, see Fig3A upper panel). Other examples of co-localization of enzyme-substrate in the same intracellular compartment include the phosphatidylinositol (PI) phosphatases Sac1, OCRL and INNP5E which meet their substrates at "discrete enzymatic locales" in the *trans*-Golgi network and the plasma membrane (Charman et al. *Traffic* 2017; PMID: 28471037, see Fig. 2B; Idevall-Hagren et al., *PNAS* 2012; PMID: 22847441; see Movie1 and 2).

5- Overall, the manuscript provides compelling evidence that PKR forms higher-order structures that share features with PB. These observations are not new, and they have been documented by others.

We would like to further clarify the novelty of our work. The crystal structure of PKR's kinase domain showing additional front-to-front PKR interfaces recently obtained by Mayo et al (Mayo et al, *Biochemistry* 2019) suggests a model of trans-phosphorylation of PKR dimers *in vivo*. The work of Corbet et al., further supports our findings on the formation of high-order PKR structures in living cells. PKR interactions with P-bodies have been reported, and indeed, the PKR clusters we describe share components with P-bodies. However, we also show that PKR clusters are autonomous entities (i.e., they do not require P-bodies to assemble), and that PKR segregate from P-body components during active de-mixing (Fig. 2B and video 3). Our results and those of Corbet et al. indicate that *in vivo* clustering is an inherent property of PKR that has not been described to date. We think our work provides new insights on the regulation of the ISR signaling by PKR and paves the way for discovering molecular mechanisms that control other ISR sensor kinases.

Minor:

1- The authors should attempt to reconcile their observation "disruption of *in vivo* PKR clustering did not suppress PKR's self-phosphorylation but rather enhanced it" with earlier reports showing the opposite *in vitro*.

We agree that our observations are in apparent contradiction with those of Mayo et al. However, we note that the report by Mayo et al (Mayo et al, Biochemistry 2019) showing the lack of trans-phosphorylation of this mutant, was based on *in vitro* experiments performed with recombinant PKR kinase domains purified from bacteria. We assessed PKR phosphorylation using a phospho-specific (T446) antibody on mammalian cell lysates. Although we cannot exclude that the mutations we introduced may change the affinity of the antibody (the immunogenic peptide used to raise the antibody encompasses the residues we mutated as indicated by the manufacturer (https://www.novusbio.com/products/pkr-antibody-sy230_nbp2-67426), the enhanced P-eIF2 α levels we observed are consistent with the notion of increased PKR activity and autophosphorylation. In our studies, we analyzed PKR containing ligand-binding domains (dsRBDs or the FKBP dimerizer-binding domain), and thus it is possible that additional interactions with ligand or other modulators, as would occur in cells, led to the differences in autophosphorylation we observed. The heterogeneous composition of PKR clusters (this work and Corbet et al. 2022) lends support to this notion. Whether additional interactions that might be influenced by PKR localization can modulate PKR's kinase activity will be the focus of future investigations and lie outside the scope of this manuscript.

2- Page 8 typo, there is an extra "corroborated" after poly I:C

We have corrected the typo in the new version of the manuscript.

Reviewer #2 (Comments to the Authors (Required)):

In this paper, Zappa et al. present data in support of a new and exciting feature of PKR signaling that may enable the control of PKR-eIF2 α interactions to fine-tune ISR signaling. The authors show that PKR forms dynamic assemblies in response to dsRNA and that these assemblies, which the authors refer to as "PKR clusters", contain phosphorylated (i.e., active) PKR. Moreover, the PKR assemblies can recruit P-body components, but they do not require P-bodies, indicating that "clustering" is an intrinsic property of active PKR. Most surprising is the finding that eIF2 α , a canonical PKR substrate, is not present in the clusters and that impeding PKR cluster assembly through mutagenesis enhances eIF2 α phosphorylation. The results support a new model in which PKR clustering may provide a mechanism to regulate PKR (and ISR signaling) by adjusting PKR-eIF2 α encounters. The data are of excellent quality, and the findings significantly increase our knowledge of the mechanisms by which PKR regulate downstream signaling. I consider this manuscript to be appropriate for the readership of JCB, and I recommend its publication. That said, I have a few minor points that need to be addressed before publication.

We thank the Reviewer for the positive comments on our work and address the minor concerns below.

Minor points:

1. One could question whether endogenous PKR would exhibit the same behavior as the ruby-tagged PKR or whether clustering is a product of a roughly 2-fold overexpression. However, interferons induce PKR during the natural physiological response to viral infection, which support the main conclusion and is likely to reflect the natural behavior of the protein. The authors could add a sentence in the discussion regarding this issue.

We agree with this observation and have added new experiments to clarify this point and corrected our narrative in the discussion in lines 82-88. Treatment of H4 cells with interferon beta (IFN β) boosted the levels of endogenous PKR to approximately 1.65-fold (Fig. S1C), which is consistent with the expression level of tagged PKR we reported in Fig. S1B. Moreover, the behavior of tagged PKR reflects that of endogenous PKR stimulated with poly I:C, as assessed by clustering ability, autophosphorylation, and signaling capacity (i.e., eIF2 α phosphorylation) (Figs. 1E, S1E, S1G).

2. The authors show that disruption of PKR clusters enhances eIF2 α phosphorylation. It will strengthen the paper if they show the enhancement of the canonical ISR downstream signaling, e.g., increased levels of ATF4.

We agree with this observation and have collected additional data showing increased levels of ATF4 upon cluster disruption in Fig. S5E. We have modified the narrative in line 263 to reflect this change.

3. The authors demonstrate that PKR clusters can form in the absence of P-bodies, but can P-bodies form in the absence of PKR?

We do not observe significant differences in P-body formation in cells in which we genetically depleted PKR, as shown below.

H4 WT and PKR KD cell lines were immunostained with Edc3. Right: quantification of the data. n>200; ns=not significant.

4. The data with the measles mutant virus convincingly show that PKR forms clusters in response to natural dsRNAs. The data of PKR clustering in mitotic cells supports the point that endogenous inputs, probably natural nuclear dsRNAs, also drive "PKR clustering". Both data sets substantiate the notion that PKR forms clusters in response to naturally occurring dsRNAs, but I find the virus infection data more compelling. The authors should consider moving the cell division data to the supplement.

We have moved the data on dividing cells to the supplement (Fig. S1J) and have made the corresponding corrections to our narrative in line 108.

Reviewer #3 (Comments to the Authors (Required)):

This is an interesting and informative study that demonstrates for the first time that PKR assembles into cytosolic clusters upon stimulation by dsRNA. These clusters are dynamic and distinct from processing bodies. Formation of these clusters is inhibited by mutations designed to interrupt a front-to-front kinase domain interaction, suggesting that this interaction is involved in cluster assembly. A model is presented where PKR clusters function to sequester PKR in a pool incapable of phosphorylating its primary substrate, eIF2alpha and thus control downstream signaling.

Overall, the work is solid and the major conclusions are supported by high quality data. However, as noted below, there are some problems with the model for the role of the clusters in attenuating substrate phosphorylation and some additional experiments are required to round out the study.

We thank the Reviewer for the suggestions to improve our manuscript and have collected additional data to round up our study. We address each concern below.

Major Issues:

1) The model where PKR clustering attenuates eIF2alpha phosphorylation has issues. It is primarily based on the observation that mNeon-eIF2alpha is excluded from the PKR clusters. These results are interpreted to indicate that the clusters act as "enzyme sinks" that regulate the extent of eIF2alpha phosphorylation and control the timing and amplitude of PKR signaling. This model begs the question of how eIF2alpha can become phosphorylated if it cannot access the active, and presumably autophosphorylated, enzyme that is sequestered in the clusters. Is eIF2alpha phosphorylation mediated by active monomeric or dimeric PKR prior to incorporation into clusters? If so, then eIF2alpha phosphorylation should precede cluster formation; however, the kinetics of eIF2alpha phosphorylation appear quite slow after addition of poly(rI:rC), with a lag-time of about 60 minutes (Fig. 5c). In contrast, cluster formation is rapid (Fig. 5B, S5A). Thus, this model is not consistent with the data.

Our model builds on the established model of PKR activation which considers that (i) monomeric PKR is inactive (Dey et al., *Cell* 2005 PMID: 16179259), (ii) dsRNA binding to PKR promotes its dimerization and autophosphorylation (Heinecke et al., 2009 *JMB* PMID: 19445956), and (iii) that back-to-back PKR dimers constitute the minimal functional entity (Dar and Sicheri *Mol Cell* 2002; PMID: 12191475). Our data expands this model by demonstrating that PKR (most likely PKR dimers) assemble onto highly-dynamic, high-molecular weight coalescences that we refer to as PKR clusters. We also found that eIF2 α is excluded from PKR clusters (Video 7; Figs. 4A, B), which suggests that enzyme and substrate do not meet in the clusters. Our model considers that cytosolic active PKR dimers freely exchange with those in the cluster, and that active PKR dimers that escape the cluster interact with and phosphorylate eIF2 α , as we have illustrated in our model figure. This model is also consistent with the kinetics of eIF2 α phosphorylation. As we mentioned above in response to the comments of Reviewer 1 (see Major point 1, part 2), three recent publications, two on the behavior of IRE1, a stress sensor kinase/RNase that clusters upon stimulation, and another one illustrating the recent findings from the laboratory of Roy Parker on PKR clustering in cells, lend additional support to our findings.

2) It is claimed that the S462A/G466L double mutant of PKR is active, and in fact it appears more active than wild type at phosphorylating eIF2alpha (Figs. 5C, 5D). However, the in vitro studies (Mayo et al., 2019) indicate that S462A PKR is partially active and G466L is essentially inactive with respect to autophosphorylation. It is possible that this discrepancy arises due to the activity of or more of the three other eIF2alpha kinases.

We agree with this observation and have provided an explanation for the discrepancy between the *in vitro* results of Mayo et al and our *in vivo* data in our response to Reviewer 1 (see Minor Point 1). The Reviewer also raises the interesting point that the other ISR kinases could play a compensatory role. Poly I:C stimulation of our mRuby-PKR double mutant cells indicated no electrophoretic mobility shift in HRI or PERK, which is consistent with a lack of activation (see figure below). Interestingly, poly I:C induced a strong upregulation of GCN2 and electrophoretic mobility shift that was accompanied by increased eIF2 α phosphorylation (lane 5 the in the figure below). We obtained a similar result upon activation of FKBP-PKR, which raises interesting possibility of interconnectivity among ISR kinases. However, given that poly I:C does not directly stimulate GCN2 (see Fig.1 in Berlanga et al., *EMBOJ* 2006; PMID 16601681) and since we did not observe any significant differences in phospho-GCN2 levels in cells expressing FKBP-PKR and FKBP-PKR^{S462A/G446L}, it is possible that a potential PKR-GCN2 crosstalk, which is not dependent on PKR's ability to cluster, exists. We will study this crosstalk in future work.

A) Western blot analysis of lysates obtained from cells expressing mRuby-PKR^{S462A/G466L} treated with different stress-inducing agents. UV: UV radiation, 90 min; Tg: 1 μ M thapsigargin, 3h; SA: 500 μ M sodium arsenite, 90 min; poly I:C 2 mg/ml, 90 min. B) Representative Western blot analysis of GCN2 phosphorylation upon activation of FKBP-PKR and FKBP-PKR^{S462A/G466L}. C) Quantification of P-GCN2 levels in cells expressing FKBP-PKR and FKBP-PKR^{S462A/G466L} treated with the dimerizer. Mean of two independent replicates. Error bars: SD.

Thus, the authors need to directly demonstrate the effect of the double mutation on the initial step in PKR activation, autophosphorylation. This is a crucial piece of data that is missing from this study. Was this experiment omitted because of the data in Figure S5B indicating that the expression level of the double mutant of mRuby-PKR is several fold lower than the wild-type?

We agree with this observation and have collected additional data that show ostensibly higher levels and faster kinetics of PKR phosphorylation in the PKR double mutant compared to the wild-type protein, despite the lower levels of the former. These results are consistent for mRuby-PKR^{S462A/G446L} and FKBP-PKR^{S462A/G446L} (Figs. S5B, C).

3) The study is predicated on the assumption that the mRuby-PKR construct accurately reflects the behavior of native PKR. The insertion of a folded mRuby domain into the unstructured linker lying between dsRBD2 and the kinase domain may affect its dsRNA binding, self-association, and enzymatic activity. Figure S1B shows that the expression levels of endogenous PKR and mRuby-PKR are close (within a factor of 2). One way to assess mRuby-PKR activity relative to the native enzyme is to compare the extent of autophosphorylation and eIF2 α phosphorylation of cells expressing either form upon stimulation with poly(I:C).

We agree with the concern of the Reviewer that tagging PKR with a fluorophore may alter its behavior. We have collected additional data comparing tagged PKR to wild-type PKR to discard any potential effects on PKR's capability for dsRNA binding, activation (i.e., phosphorylation) kinetics, downstream signal transduction (i.e., eIF2 α phosphorylation) and ability to cluster. The new data are shown in Figs. 1E, S1E, and S1G. We have also adjusted our narrative in lines 81-88 and 93-96 to include these new analyses.

4) The present work cannot distinguish whether PKR cluster formation requires autophosphorylation or can be driven solely by assembly on dsRNA. A simple way to assess this is to compare cluster formation in cells expressing either wild-type mRuby-PKR or an inactive mutant of mRuby-PKR (e.g., K296R). This experiment would greatly enhance our understanding of the clustering process.

The model of activation of other stress sensor kinases that form high-order dynamic assemblies (e.g., the ER stress sensors IRE1 and PERK) suggests that unfolded protein ligand binding is a pre-requisite for formation of back-to-back dimers and activation (i.e., autophosphorylation). It is likely that PKR follows a similar mechanism wherein dsRNA binding provides the means to "bridge" inactive monomers into active back-to-back dimers. This notion is supported by data showing that mutations in mRuby-PKR and FKBP-PKR that abrogate PKR's enzymatic activity (K296R and T446A mutants), do not abrogate PKR's clustering ability but rather modestly enhance it (Figs. 3H, S3C and Video 6), which suggests that autophosphorylation counters cluster formation. The same results were obtained by Corbet et al. (<https://doi.org/10.1101/2022.01.14.476399>). Thus, enzymatic activity is not required for PKR cluster assembly.

Minor Issues:

1) page 4: The statement describing PKR clusters as "dynamic coalescences that are reminiscent of coacervates" is vague. Do the authors believe that the clusters represent liquid-liquid phase separated coacervates? Alberti et al. (2019) point out several caveats regarding the use of 1,6 hexanediol and FRAP as criteria for a liquid-liquid phase separation.

We agree with the observations of Simone Alberti on the caveats of using hexanediol as an inclusion criterion for defining liquid-liquid phase separation. The dynamic behavior of the PKR clusters we observed is reminiscent of entities that engage in liquid-liquid phase separation. For instance, we note that PKR clusters interact P-bodies, which are liquid-like components, which suggests shared common biophysical principles. However, we acknowledge we do not have sufficient biophysical data to formally claim that PKR clusters are liquid-liquid phase separated coacervates. For these reasons, we have not delved into a detailed description of the biophysical nature of PKR clusters other than mentioning they resemble coacervates.

2) Figure 1D: It is interesting that PKR phosphorylation appears to occur only in the clusters and the bulk of the mRuby-PKR in the cytoplasm does not appear to be phosphorylated. This appears to support a model where cluster formation requires autophosphorylation and not just assembly on dsRNA (See major issue #4).

We have addressed the issue of cytosolic vs. cluster-bound PKR in our response to Reviewer 1 (see Major point 1, part 2).

3) *Figure 1F: The Y-axis labels on the right side have a typo (change 1.0 to 10.0). How were these data analyzed to determine that 45% of the cells formed clusters?*

We have corrected the typo in the axis. Regarding our analysis, we counted the number of cells in which we observed clusters and expressed this number as the percent of the total number of cells expressing GFP (infected cells).

4) *Figure 1G: The cluster formation in cells undergoing mitosis is not very convincing. The intensities of the clusters are barely above the background fluorescence (soluble monomeric mRuby-PKR?). In any case, the statement in the discussion (page 9) that this data indicates that cluster formation is driven by endogenous dsRNAs is speculative. Factors other than the release of nuclear dsRNAs could drive cluster formation during cell division.*

We based our conclusion of PKR cluster formation in dividing cells on the previous reports of Kim et al. (Kim et al., *Genes Dev* 2004; PMID: 24939934 and Kim et al., *Mol Cell* 2018; PMID: 30174290) showing activation of PKR during mitosis. One conclusion of these studies was that the leakage of nuclear dsRNA (e.g., snoRNAs, see Youssef et al., *PNAS* 2015; PMID: 25848059) may drive PKR's activation. However, we agree that our data on PKR cluster formation in dividing cells data is less compelling. This point was also raised by Reviewer 2 and for this reason we have moved our data on dividing cells to the supplement (Fig. S1J).

5) *Figure 1H: The Y-axis labels at the breakpoint are incorrect. Also, the maximum for a normalized intensity should be 1, not 100.*

We have corrected this issue.

6) *Page 4: The rate of exchange between clusters and cytosolic pools of mRuby-PKR was measured from FRAP and is stated as 3.93 s. First, this is a half-life, not a rate. Second, the distribution of the half-lives (Fig. S1E) is so broad that an average half-life is not meaningful.*

We thank the reviewer for pointing this out and have corrected our narrative to indicate the measurement of half-life, not rate.

7) *Figure S1: In the caption "F" is labeled as "E".*

We have corrected this issue.

8) *Figure S1B: The abbreviation PKR KD is meant to refer to knockdown but can also be interpreted as kinase domain.*

We have corrected this issue.

9) *Page 3: It is stated that the size ranges from 0.22 to 5 um in diameter but Fig. S1D indicates that the size range is very large and goes up to ~8 um.*

We have corrected this issue and indicated that the size ranges from 0.22 to 8 um in line 91.

10) *Page 5: Reference to S2A and S2B in the text should be S2.1A and S2.1B.*

We have corrected this issue.

11) Page 5: *It is stated that ejection of mRuby-PKR from GFP-Dcp1a-containing clusters occurs in about 15 minutes yet the images in Fig. 2B only begin at 25 minutes.*

The ejection of mRuby-PKR from GFP-Dcp1a-containing clusters occurs 15 minutes *after* the formation of the clusters, which occurs at ~25 minutes after poly I:C transfection. We have clarified this in the narrative in lines 150-153.

12) Page 5: *The origin of the discrepancy between the extent of colocalization of PB and PKR clusters determined by fixed and live cell imaging should be clarified.*

This discrepancy is the product of lack of temporal resolution and fixation in immunofluorescence analyses. Moreover, fixation can generate colocalization artifacts by collapsing diffusible juxtaposed structures, such as PKR clusters and PBs.

13) Page 6: *The extent of depletion of PBs by CHX and knockdown of 4E-T is not clear in Fig. S2.1C. Some quantitation is warranted.*

We have added the quantification of those data to Fig S2.1D.

14) Figure 2D: *How can the violin plots exceed the value of 100 when the scale is % maximum?*

The violin plots are smoothed out by the software we used to prepare the (GraphPad prism) which artificially extends the tails of the distribution rather than truncate them at 0 or 100.

15) Page 6: *The reference to Lee et al, 2020 should be 2020b.*

We have corrected this issue.

16) Page 7, line 2: *Replace RBDs with dsRBDs.*

We have corrected this issue.

17) Figure 4B: *It appears that the phospho-eIF2alpha staining is prominent in the nucleus in the steady-state image. This protein should be primarily cytosolic.*

We agree that eIF2a is primarily cytosolic. The faint nuclear staining observed in the steady-state image is antibody background noise due to the low level of the specific antigen (P-eIF2 α) in steady state conditions.

18) Figure 5A: *The effects of the mutations on higher-order assembly of PKR are not well represented by the cartoon. The WT representation implies that the interaction involves the phosphorylated residue(s) in the activation loop. However, the crystal structures exhibiting a front-to-front interface were obtained for the unphosphorylated PKR kinase so this interaction cannot require autophosphorylation. The significance of the two different arrows on the right (mutant) side of the figure is unclear. Does it indicate that there is still some sort of front-to-front interaction in the presence of the two mutations? The data in Figure 5B argues that the interaction is pretty well gone.*

The data showing that the catalytically dead mutants PKR^{K296R} and PKR^{T446A} form clusters (Figs. 3H, S3C, and video 6 in the revised manuscript; See also Corbet et al., fig. 4G) suggest that the FTF interactions are destabilized by phosphorylation and that phosphorylation is not required for the assembly of high-order PKR assemblies. Our data obtained with the PKR^{S462A/G462L} mutant indicates that the destabilization of the FTF interfaces between PKR's catalytic c-lobes that leads to disrupted clustering does not diminish PKR autophosphorylation (Figs. S5B, C). While we cannot exclude that phospho-transfer could occur *in trans* among preassembled dimers, it is possible that mutations in the FTF interfaces are not sufficient to abrogate transient interactions between PKR molecules necessary for phospho-transfer.

19) *Figure S5B: The data for wild-type mRuby-PKR cluster formation does not agree with Fig. 1B. The number of clusters per cell is much less and the kinetics are much slower. Why?*

The data agree with one another, but in Fig S5B we quantified the absolute maximum number of clusters per cell. This analysis was performed in Fiji (V2.3) and the plot in Fig. 1B was prepared by quantifying the average number of clusters over time using an automated pipeline we have designed on Cell Profiler as described in the Materials and Methods section.

20) *Figure 5E: It is hard to see by eye that there is any difference in the staining of the cells expressing WT and mutant FKBP-PKR. At the resolution and magnification of the images both seem to show clusters. Thus, the dramatic effect in the quantitation shown in Figure 5F is suspect.*

We have cropped the figures in figure 5E. We also include single-cell crops to substantiate our claims.

21) *Figure 5: There are two part D and part E descriptions in the legend.*

We have corrected this issue.

22) *Figure S5C is not described fully and there are multiple PKR bands in the p-PKR immunoblot. It does not clearly demonstrate enhanced autophosphorylation of the double mutant of FKBP-PKR. If it did show this, it would contradict Figure 5G.*

We have collected higher-quality IP-WB data to substantiate our claims. The new data are in Fig. S5B and S5C.

23) *The reference for Belyy et al. (2021) is incorrect.*

We have corrected this issue.

24) *There is no Figure S4.*

We do not have any supplementary data for figure 4; any pointers to a supplemental figure 4 have been removed.

Warmest Regards,

Francesca and Diego

March 25, 2022

RE: JCB Manuscript #202111100R

Dr. Diego Acosta-Alvear
University of California Santa Barbara
Molecular, Cellular and Developmental Biology
2115 Life Sciences Building
Santa Barbara 93106

Dear Dr. Acosta-Alvear:

Thank you for submitting your revised manuscript entitled "Signaling by the integrated stress response kinase PKR is fine-tuned by dynamic clustering". We would be happy to publish your paper in JCB pending final revisions necessary to meet our formatting guidelines (see details below). In view of comments by Reviewer 3, please also adjust the discussion of your proposed model and conclusions to acknowledge caveats and alternative interpretations of your observations.

A. MANUSCRIPT ORGANIZATION AND FORMATTING:

Full guidelines are available on our Instructions for Authors page, <https://jcb.rupress.org/submission-guidelines#revised>. Submission of a paper that does not conform to JCB guidelines will delay the acceptance of your manuscript.

1) Text limits: Character count for Articles is < 40,000, not including spaces. Count includes abstract, introduction, results, discussion, and acknowledgments. Count does not include title page, figure legends, materials and methods, references, tables, or supplemental legends.

2) Figures limits: Articles may have up to 10 main text figures.

3) Figure formatting: Scale bars must be present on all microscopy images, including inset magnifications. Molecular weight or nucleic acid size markers must be included on all gel electrophoresis.

****Please add a scale bar to Fig 1A, Supp Fig 2.2A (last panel), and 2.2B. Please ensure scale bars in Fig 3D and other images are visible.**

4) Statistical analysis: Error bars on graphic representations of numerical data must be clearly described in the figure legend. The number of independent data points (n) represented in a graph must be indicated in the legend. Statistical methods should be explained in full in the materials and methods. For figures presenting pooled data the statistical measure should be defined in the figure legends. Please also be sure to indicate the statistical tests used in each of your experiments (either in the figure legend itself or in a separate methods section) as well as the parameters of the test (for example, if you ran a t-test, please indicate if it was one- or two-sided, etc.). Also, if you used parametric tests, please indicate if the data distribution was tested for normality (and if so, how). If not, you must state something to the effect that "Data distribution was assumed to be normal but this was not formally tested."

****Please indicate the number of biological and technical replicates analyzed in Fig 5F and Supp Fig 1F. Also indicate the parameters for all Students t-tests, and the type of test performed in Fig 4C.**

5) Abstract and title: The abstract should be no longer than 160 words and should communicate the significance of the paper for a general audience. The title should be less than 100 characters including spaces. Make the title concise but accessible to a general readership.

6) Materials and methods: Should be comprehensive and not simply reference a previous publication for details on how an experiment was performed. Please provide full descriptions in the text for readers who may not have access to referenced manuscripts.

**** Please describe the methodology used for CRISPR interference and FRAP analysis, in lieu of or in addition to the associated references.**

7) Please be sure to provide the sequences for all of your primers/oligos and RNAi constructs in the materials and methods. You must also indicate in the methods the source, species, and catalog numbers (where appropriate) for all of your antibodies.

Please also indicate the acquisition and quantification methods for immunoblotting/western blots.

8) Microscope image acquisition: The following information must be provided about the acquisition and processing of images:

- a. Make and model of microscope
- b. Type, magnification, and numerical aperture of the objective lenses
- c. Temperature
- d. Imaging medium
- e. Fluorochromes
- f. Camera make and model
- g. Acquisition software
- h. Any software used for image processing subsequent to data acquisition. Please include details and types of operations involved (e.g., type of deconvolution, 3D reconstitutions, surface or volume rendering, gamma adjustments, etc.).

**Please include objective type (PLAN-APO, PLAN-FLUOR, etc) and imaging medium.

10) Supplemental materials: There are strict limits on the allowable amount of supplemental data. Articles may have up to 5 supplemental figures. Please also note that tables, like figures, should be provided as individual, editable files. A summary of all supplemental material should appear at the end of the Materials and methods section.

**Please relabel supplemental figures in order 1-5.

13) ORCID IDs: ORCID IDs are unique identifiers allowing researchers to create a record of their various scholarly contributions in a single place. At resubmission of your final files, please consider providing an ORCID ID for as many contributing authors as possible.

WHEN APPROPRIATE: The source code for all custom computational methods published in JCB must be made freely available as supplemental material hosted at www.jcb.org. Please contact the JCB Editorial Office to find out how to submit your custom macros, code for custom algorithms, etc. Generally, these are provided as raw code in a .txt file or as other file types in a .zip file. Please also include a one-sentence summary of each file in the Online Supplemental Material paragraph of your manuscript.

Please note that JCB now requires authors to submit Source Data used to generate figures containing gels and Western blots with all revised manuscripts. This Source Data consists of fully uncropped and unprocessed images for each gel/blot displayed in the main and supplemental figures. Since your paper includes cropped gel and/or blot images, please be sure to provide one Source Data file for each figure that contains gels and/or blots along with your revised manuscript files. File names for Source Data figures should be alphanumeric without any spaces or special characters (i.e., SourceDataF#, where F# refers to the associated main figure number or SourceDataFS# for those associated with Supplementary figures). The lanes of the gels/blots should be labeled as they are in the associated figure, the place where cropping was applied should be marked (with a box), and molecular weight/size standards should be labeled wherever possible.

B. FINAL FILES:

Thank you for this interesting contribution, we look forward to publishing your paper in Journal of Cell Biology.

Sincerely,

Judith Frydman
Monitoring Editor
Journal of Cell Biology

Tim Fessenden
Scientific Editor
Journal of Cell Biology

Reviewer #1 (Comments to the Authors (Required)):

The authors addressed my main concerns, and they made some convincing arguments regarding their interpretations of the data. They also carried out additional experiments, such as those with untagged PKR, to make the paper much better. Given the overall enthusiasm from the other two reviewers, I support the publication of the manuscript.

Reviewer #2 (Comments to the Authors (Required)):

The authors have addressed many of the criticisms of the 3 reviewers. However, a large number of questions remain unanswered. Given the novelty of the findings, the complexity of the system, and the recent posting of a paper by Corbett et al., to BioRxiv, which supports the authors' conclusions, I recommend publication.

Reviewer #3 (Comments to the Authors (Required)):

The changes incorporated into the revised version of this manuscript address several of the concerns raised in my review and those of the other reviewers. The revised manuscript is substantially improved. However, there are two areas that remain

problematic.

1. The authors still oversell their model that ascribes functional significance to the formation of PKR clusters. It should be stated as a hypothesis rather than asserting it as a solid conclusion. For example, it is stated on lines 68-70, "Taken together, our results highlight an unexpected feature of the ISR in which compartmentalization modulates PKR-eIF2alpha interactions to fine-tune signaling." The model, summarized in the abstract, states, "PKR clustering limits encounters between PKR and eIF2alpha to buffer downstream signaling and prevent the ISR from misfiring." This model rests primary on two pieces of data: 1) eIF2alpha is excluded from the clusters, and 2) introduction of two PKR mutation designed to disrupt the front-to-front interface reduces cluster formation and enhances PKR activity (autophosphorylation and eIF2alpha phosphorylation). The latter observations indicate correlation, not causation. There is no evidence that the enhanced PKR activity is actually caused by the disruption of the clusters. Maybe it is a side-effect. The observation cited in the paper and the rebuttal that IRE1 also forms clusters that exclude substrate is interesting as an analogy to PKR but does not directly speak to any functional role for the PKR clusters as limiting access to the substrate. In the rebuttal, the authors state that only 5% of the PKR is recruited into clusters. That would mean that sequestering PKR into substrate-inaccessible clusters would at most result in a 5% reduction in eIF2alpha phosphorylation. That is very small signal change that may not be biologically relevant. Can the authors exclude an admittedly boring model where formation of the clusters is simply a small side-reaction associated with PKR activation but has no regulatory significance?

2. The fact that this PKR double mutant shows any activity in cells remains troubling. As mentioned in my original review, Mayo et al. (2019) reported that S462A PKR is partially active and G466L is completely inactive in vitro with respect to autophosphorylation. The explanations offered in the rebuttal to my concerns and those of reviewer 1 to explain this discrepancy are not convincing. Specifically, the in vitro activity data were obtained with full-length PKR, not isolated kinase domain. From a structural point of view, it is hard to understand how PKR would be capable of undergoing autophosphorylation in the absence of a front-to-front interaction. This domain-swapped interface is believed to mediate the critical trans-autophosphorylation of T446 within the activation loop. How can PKR undergo autophosphorylation if this reaction cannot occur?

Dear Judith and Tim,

We are very pleased about the enthusiastic support for the publication of our study in JCB and we thank you and the reviewers for your thoughtful comments and critiques. After careful consideration, we have also addressed the concerns raised by Reviewer #3 and have made changes to the discussion of the manuscript. In addition, we have also uploaded a cover image for your consideration.

We hope that you will find the revised manuscript suitable for publication in JCB. Thank you,

Best wishes

Francesca and Diego

Reviewer #1 (Comments to the Authors (Required)):

The authors addressed my main concerns, and they made some convincing arguments regarding their interpretations of the data. They also carried out additional experiments, such as those with untagged PKR, to make the paper much better. Given the overall enthusiasm from the other two reviewers, I support the publication of the manuscript.

We would like to thank Reviewer#1 for the thoughtful and constructive critique of our manuscript. We are glad that the Reviewer found that we have satisfactorily addressed all his/her concerns.

Reviewer #2 (Comments to the Authors (Required)):

The authors have addressed many of the criticisms of the 3 reviewers. However, a large number of questions remain unanswered. Given the novelty of the findings, the complexity of the system, and the recent posting of a paper by Corbett et al., to BioRxiv, which supports the authors' conclusions, I recommend publication.

We would like to thank Reviewer #2 for recommending the publication of our work and agree that our study opens multiple questions which we will address in subsequent work.

Reviewer #3 (Comments to the Authors (Required)):

The changes incorporated into the revised version of this manuscript address several of the concerns raised in my review and those of the other reviewers. The revised manuscript is substantially improved. However, there are two areas that remain problematic.
1. The authors still oversell their model that that ascribes functional significance to the formation of PKR clusters. It should be stated as a hypothesis rather than asserting it as a solid conclusion. For example, it is stated on lines 68-70, "Taken together, our results highlight an unexpected feature of the ISR in which compartmentalization modulates PKR-eIF2alpha interactions to fine-tune signaling." The model, summarized in the abstract, states, "PKR clustering limits encounters between PKR and eIF2alpha to buffer downstream signaling and prevent the ISR from misfiring." This model rests primary on two pieces of data: 1) eIF2alpha is excluded from the clusters, and 2) introduction of two PKR mutation designed to disrupt the front-to-front interface reduces cluster formation and enhances PKR activity (autophosphorylation and eIF2alpha phosphorylation). The latter observations indicate correlation, not causation. There is no evidence that the enhanced PKR activity is actually caused by the disruption of the clusters. Maybe it is a side-effect. The observation cited in the paper and the rebuttal that IRE1 also forms clusters that exclude substrate is interesting as an analogy to PKR but does not directly speak to any functional role for the PKR clusters as limiting access to the substrate. In the rebuttal, the authors state that only 5% of the PKR is recruited into clusters. That would mean that sequestering PKR into substrate-inaccessible clusters would at most result in a 5% reduction in eIF2alpha phosphorylation. That is

very small signal change that may not be biologically relevant. Can the authors exclude an admittedly boring model where formation of the clusters is simply a small side-reaction associated with PKR activation but has no regulatory significance?

We thank the Reviewer for the suggestion. As requested, we have now de-emphasized some of our conclusions and have modified the manuscript accordingly (see lines 399-408). At the same time, our interpretation builds on the notion that functional compartmentalization of kinases and stress sensor proteins is a fundamental aspect of signaling (PMIDs 19079237; 20798350; 10854322; 23084743; 20208517 and <https://doi.org/10.1101/2022.01.14.476399>).

The Reviewer also raised the concern that only a fraction of PKR is recruited into the clusters. It is noteworthy that PKR levels (~25-75 nanomolar) are approximately 100,000 times lower than those of its substrate eIF2 α (~2.5 millimolar) (<https://doi.org/10.1101/2022.01.01.474691>). Considering this observation alongside the enzyme turnover number of kinases, and the facts that a) signaling pathways are non-linear and b) that only a small fraction of eIF2-P is sufficient to inhibit general translation (eIF2 is more abundant than eIF2B; PMID: 29735639), a small modulation in PKR activity should impact downstream signaling. Consistent with this idea, we found that disruption of PKR clusters through mutagenesis (PKR^{S462A/G466L} mutation) led to a measurable increase in eIF2-P (Fig. 5D). A comprehensive characterization of how PKR clustering regulates cell and organismal function will be carried out in future studies.

2. The fact that this PKR double mutant shows any activity in cells remains troubling. As mentioned in my original review, Mayo et al. (2019) reported that S462A PKR is partially active and G466L is completely inactive in vitro with respect to autophosphorylation. The explanations offered in the rebuttal to my concerns and those of reviewer 1 to explain this discrepancy are not convincing. Specifically, the in vitro activity data were obtained with full-length PKR, not isolated kinase domain. From a structural point of view, it is hard to understand how PKR would be capable of undergoing autophosphorylation in the absence of a front-to-front interaction. This domain-swapped interface is believed to mediate the critical trans-autophosphorylation of T446 within the activation loop. How can PKR undergo autophosphorylation if this reaction cannot occur?

We thank the reviewer for this important comment. The Reviewer is correct; full-length PKR was used in Mayo et al. *in vitro* PKR phosphorylation assays while PKR's kinase domain was used in their crystallization studies. We apologize for this overlook in our rebuttal letter. That said, it is important to consider that in cells molecular interactions and post-translational modifications (PTMs) are crucial for signaling, and these processes are unlikely to be replicated in a test tube containing only recombinant PKR +/- poly I:C. In future studies, we will examine how heterologous interactions and PTMs impact PKR activation and cluster assembly.